# Association Graph Learning for Multi-Task Classification with Category Shifts

**Jiayi Shen**[1], **Zehao Xiao**[1], **Xiantong Zhen**[1,2] *, **Cees G. M. Snoek**[1], **Marcel Worring**[1]

[1] AIM Lab, University of Amsterdam, Netherlands,
{j.shen, z.xiao, C.G.M.Snoek, m.worring}@uva.nl
[2] Inception Institute of Artificial Intelligence, Abu Dhabi, UAE, zhenxt@gmail.com

## Abstract

In this paper, we focus on multi-task classification, where related classification tasks share the same label space and are learned simultaneously. In particular, we tackle a new setting, which is more realistic than currently addressed in the literature, where categories shift from training to test data. Hence, individual tasks do not contain complete training data for the categories in the test set. To generalize to such test data, it is crucial for individual tasks to leverage knowledge from related tasks. To this end, we propose learning an association graph to transfer knowledge among tasks for missing classes. We construct the association graph with nodes representing tasks, classes and instances, and encode the relationships among the nodes in the edges to guide their mutual knowledge transfer. By message passing on the association graph, our model enhances the categorical information of each instance, making it more discriminative. To avoid spurious correlations between task and class nodes in the graph, we introduce an assignment entropy maximization that encourages each class node to balance its edge weights. This enables all tasks to fully utilize the categorical information from related tasks. An extensive evaluation on three general benchmarks and a medical dataset for skin lesion classification reveals that our method consistently performs better than representative baselines.[1]

## 1 Introduction

Multi-task learning aims to simultaneously solve several related tasks [9, 69] by sharing information and has attracted much attention in recent years. In this paper, we focus on multi-task classification under the multi-input multi-output setting [70, 37, 46, 69]. In this setting, the label space is shared but each task operates on a different type of visual modality or the same modality collected from different environments or equipment. As a consequence, each task follows different data distributions for the shared labels. The intuition behind multi-task classification is that tasks having the same label space provide partial knowledge of the distributions that can be shared among all tasks to reach a better view of the full distribution, which in turn benefits the individual tasks.

A real-world challenge for multi-task classification is category shift, where the categories in the testing phase are shared but during training not all classes are present for each individual task. This challenge is common in various realistic scenarios, such as skin lesion classification [31, 64], fault diagnosis [56, 14], or remote sensing scene classification [38]. For example, in skin lesion classification, data provided by different hospitals or healthcare facilities should lead to the same set of diagnoses [31, 64]. Unfortunately, due to patient populations or proprietary use regulations, these tasks do not share

---

*Currently with United Imaging Healthcare, Co., Ltd., China.

[1]Code: https://github.com/autumn9999/MTC-with-Category-Shifts.git

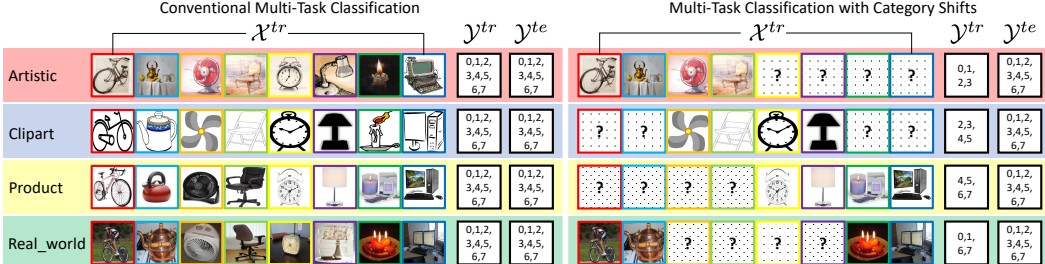

Figure 1: Comparison between multi-task classification without (left) and with category shifts. $\mathcal{X}^{tr}$ denotes the training set. Each row and column of the training data corresponds to one task and one category. Each visual modality (e.g., Artistic) corresponds to one task. $\mathcal{Y}^{tr}$ and $\mathcal{Y}^{te}$ denote the label spaces in training and test phases, respectively. We address the category shifts in multi-task classification, where instances from several categories are unavailable during individual task training.

the same diagnosis categories at training time. For example, some institutions miss instances from *melanoma* and *basal cell carcinoma* while others lack *dermatofibroma* and *benign keratosis* [64]. In this case, it is beneficial to expand the diagnostic scope of different hospitals or healthcare facilities by simultaneously learning their training data and improving the overall prediction accuracy. Motivated by these realistic scenarios, we propose a new multi-task setting, namely multi-task classification with category shifts. Figure 1 shows comparisons between multi-task classification with and without category shifts. The goal of the proposed setting is to explore task-relatedness with an incomplete training label space to improve the generalization ability of the categorical information at test time.

To deal with the category shifts, we propose to learn an association graph to transfer knowledge among tasks. The association graph is constructed over three different types of nodes: task, class and instance nodes. To model the complex relationships among these heterogeneous nodes, we apply different learnable metric functions to construct the edge weights among different types of nodes. To propagate knowledge between the nodes, we apply message passing to update each node in the association graph according to their relationships. Essentially, the association graph stores the task and class-specific knowledge during training time and enhances each instance feature by associating it with other task and class nodes during inference. In the constructed association graph, the relationships between class and task nodes tend to be biased due to the category shifts. This hinders tasks in utilizing the categorical information for missing classes. To avoid these spurious correlations between task and class nodes, we introduce assignment entropy maximization. This regularization of the association graph encourages each class node to balance its edge weights with all tasks, enabling them to fully utilize the categorical information.

We evaluate our model on three multi-task classification benchmarks and a medical dataset for skin lesion analysis to demonstrate that the proposed model performs better under category shifts. We also provide detailed analyses to show how the proposed association graph enhances the categorical information of each instance with transferred knowledge.

## 2 Problem Statement

We first formally introduce the new problem setting of multi-task classification with category shifts. We consider $T$ related classification tasks $\{\mathcal{D}_t\}_{t=1}^{T}$. Each task contains a training set $\mathcal{D}_t^{tr}$ and a test set $\mathcal{D}_t^{te}$. We define $\mathcal{D}_t^{tr}=\{\mathcal{X}_t^{tr}, \mathcal{Y}_t^{tr}\}$, where $\mathcal{X}_t^{tr}$ denotes the set of training data from the $t$-th task and $\mathcal{Y}_t^{tr}$ is the corresponding label space in the training phase and $t \in \{1, 2, ..., T\}$. Likewise, we define $\mathcal{D}_t^{te}=\{\mathcal{X}_t^{te}, \mathcal{Y}_t^{te}\}$. In addition, we define the entire label space of all tasks as $\mathcal{Y}$. The conventional multi-task classification setting, e.g., [37, 46], is a specific instantiation of our setting where all tasks share the entire label space at both training and test time $\mathcal{Y}_t^{tr}=\mathcal{Y}_t^{te}=\mathcal{Y}$.

**Definition 1** (Category Shifts in Multi-Task Classification). *For each task, the training label space is a proper subset of the test label space $\mathcal{Y}_t^{tr} \subset \mathcal{Y}_t^{te}$, where $t \in \{1, 2, .., T\}$. The union of the training label spaces of all tasks $\mathcal{Y}=\bigcup_{t=1}^{T} \mathcal{Y}_t^{tr}$ determines the label space for the test phase.*

We provide a visual illustration of the difference between multi-task classification without and with category shifts in Figure 1. The goal of the proposed setting is to explore task-relatedness in the presence of missing classes to improve the generalization ability of categorical information. To

study the impact of different degrees of category shifts, we introduce the missing rate $\gamma$ that formally measures the degree of category shifts, with higher missing rates yielding more severe category shifts, and therefore more challenging conditions for each task.

**Definition 2** (Missing Rate). *Given $T$ related classification tasks with entire label space $\mathcal{Y}$, the missing rate $\gamma$ is the average rate of the number of missing classes with respect to the size of the entire label space, $\gamma = \frac{1}{T} \sum_{t=1}^{T} \left( \frac{|\mathcal{Y}| - |\mathcal{Y}_t^{tr}|}{|\mathcal{Y}|} \right)$.*

Having defined the problem setting, we are now ready to present the first multi-task classification method tailored to handle category shifts.

## 3 Methodology

### 3.1 Learning the association graph for knowledge transfer

To deal with the category shifts, we propose to learn an association graph to transfer knowledge among tasks for each class. The necessity of designing the graph is due to the category shifts requiring knowledge transfer among tasks and classes, which varies across different classes. We construct an undirected graph over three types of nodes: task, class and instance nodes. The association graph stores the task and class-specific knowledge from the training data in the task and class nodes. The edges encode the relationships between the nodes, enabling the relevant knowledge to be transferred to each instance node. Since the nodes are heterogeneous, we apply different learnable metric functions to compute edge weights for different types of nodes. To better explain the construction, we provide an illustration of the association graph in Figure 2.

The rationale behind our model is that relevant knowledge is transferred among nodes in the association graph to enhance the categorical information of each instance, making the instance more discriminative. During training, the model learns the ability to update all nodes in the graph by transferring the relevant knowledge. At inference time, the ability is generalized to each test instance of either observed or missing categories. Thus, each test instance is refined with the relevant knowledge stored in the association graph, which reduces the category shifts from the training to the test set. For clarity, we introduce the main components of the association graph: task and class graphs.

**Task graph.** Given $T$ related tasks, we introduce the task graph to model the relationships between tasks. The features of the nodes in the task graph are task-specific representations, each of which aggregates all features from the corresponding task. We define the node of the $t$-th task as follows:

$$\mathbf{v}_t = \frac{1}{N_t^{tr}} \sum_{i=1}^{N_t^{tr}} \mathcal{E}(\mathbf{x}_i), \tag{1}$$

where $\mathbf{x}_i$ is a training instance belonging to the $t$-th task and $N_t^{tr}$ is the number of training instances from the corresponding task. $\mathcal{E}$ is a feature extractor shared by all tasks, which embeds each instance into a $d$-dimensional feature space. $\mathcal{E}$ is a feature extractor shared by all tasks, which embeds each instance into a $d$-dimensional feature space.

With the task nodes, we further define the edges between the task nodes $\mathbf{v}_i$ and $\mathbf{v}_j$ as $A_{\mathcal{T}}(\mathbf{v}_i, \mathbf{v}_j)$. The weight of the edge is determined by the learnable similarity between the task nodes, which is formulated as follows:

$$A_{\mathcal{T}}(\mathbf{v}_i, \mathbf{v}_j) = \sigma(\mathbf{W}_{\mathcal{T}}(|\mathbf{v}_i - \mathbf{v}_j|/\alpha_{\mathcal{T}}) + \mathbf{b}_{\mathcal{T}}), \tag{2}$$

where $\mathbf{W}_{\mathcal{T}}$ and $\mathbf{b}_{\mathcal{T}}$ are the learnable parameters for the task graph. $\alpha_{\mathcal{T}}$ is a scalar and $\sigma$ is the sigmoid function used to normalize the edge weight between 0 and 1. The edge weight indicates the proximity between the $i$-th and $j$-th tasks. We denote the task graph as $\mathcal{G}_{\mathcal{T}} = (\mathbf{V}_{\mathcal{T}}, \mathbf{A}_{\mathcal{T}})$. In the task graph, $\mathbf{V}_{\mathcal{T}} = \{\mathbf{v}_t | t \in [1, T]\} \in \mathbb{R}^{T \times d}$ is the set of all task nodes and $\mathbf{A}_{\mathcal{T}} = \{A_{\mathcal{T}}(\mathbf{v}_i, \mathbf{v}_j) | i, j \in [1, T]\} \in \mathbb{R}^{T \times T}$ is the corresponding adjacency matrix, which characterizes task relationships.

**Class graph.** Likewise, we define the class graph as $\mathcal{G}_{\mathcal{C}} = (\mathbf{V}_{\mathcal{C}}, \mathbf{A}_{\mathcal{C}})$, where each node represents the corresponding categorical information. Here $\mathbf{V}_{\mathcal{C}} = \{\mathbf{k}_c | c \in [1, C]\} \in \mathbb{R}^{C \times d}$, where $C$ denotes

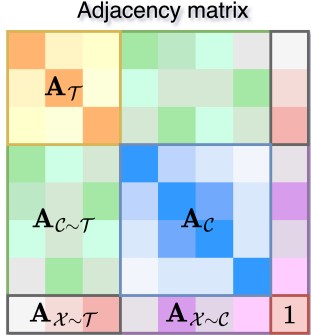

Figure 2: The illustrative association graph (left) and the adjacency matrix (right). The association graph contains the task graph, the class graph and an instance node. Nodes in the task and class graphs store the task and class-specific knowledge during training. Edges in the association graph encode the various relationships among nodes, facilitating knowledge transfer between them. Single and double lines denote the edges among the homogeneous nodes and the MULTIPLE edges among heterogeneous nodes, respectively.

the size of the entire label space of all tasks. We define the features of the node of the $c$-th class as:

$$\mathbf{k}_c = \frac{1}{N_c^{tr}} \sum_{m=1}^{N_c^{tr}} \mathcal{E}(\mathbf{x}_m), \tag{3}$$

where $\mathbf{x}_m$ is a training instance from $c$-th class and $N_c^{tr}$ is the number of training instances of the corresponding class. The edge weight between class nodes $A_{\mathcal{C}}(\mathbf{k}_i, \mathbf{k}_j)$ is formulated as:

$$A_{\mathcal{C}}(\mathbf{k}_i, \mathbf{k}_j) = \sigma(\mathbf{W}_{\mathcal{C}}(|\mathbf{k}_i - \mathbf{k}_j|/\alpha_{\mathcal{C}} + \mathbf{b}_{\mathcal{C}})), \tag{4}$$

where $\mathbf{W}_{\mathcal{C}}$ and $\mathbf{b}_{\mathcal{C}}$ are the learnable parameters for the class graph. $\alpha_{\mathcal{C}}$ is a fixed scalar. The motivation for designing the metric function with learnable parameters is to explore the task-specific or class-specific semantic information in the corresponding graph. The adjacency matrix of the class graph is $\mathbf{A}_{\mathcal{C}} = \{A_{\mathcal{C}}(\mathbf{k}_i, \mathbf{k}_j) | i, j \in [1, C]\} \in \mathbb{R}^{C \times C}$.

**Association graph.** Having defined the task and class graphs, we build the association graph by connecting both graphs with each instance node, as shown in Figure 2. We use the association graph to enhance instance feature learning by leveraging task representations (nodes in the task graph) and categorical information (nodes in the class graph). To do so, we query each instance in the task and class graphs. Formally, we define an instance node as $\mathbf{V}_{\mathcal{X}} = \{\mathcal{E}(\mathbf{x})\} \in \mathbb{R}^{1 \times d}$.

We build up the connection of the instance node to each node of the task graph and the class graph, which gives the instance node access to the knowledge stored in the task and class graph. The edge between the instance node $\mathcal{E}(\mathbf{x})$ and a task node $\mathbf{v}_t$ is denoted by $A_{\mathcal{X} \sim \mathcal{T}}(\mathbf{x}, \mathbf{v}_t)$. The weight of the edge is obtained by the normalized similarity between the instance and task node, $A_{\mathcal{X} \sim \mathcal{T}} = \text{softmax}(\frac{\mathcal{E}(\mathbf{x})^{\top} \mathbf{v}_t}{\sqrt{d}})$. The corresponding adjacency matrix is $\mathbf{A}_{\mathcal{X} \sim \mathcal{T}} = \{A_{\mathcal{X} \sim \mathcal{T}}(\mathbf{x}, \mathbf{v}_t) | t \in [1, T]\} \in \mathbb{R}^{1 \times T}$. Likewise, we define the adjacency matrix between instance nodes and the class graph as $\mathbf{A}_{\mathcal{X} \sim \mathcal{C}} = \{A_{\mathcal{X} \sim \mathcal{C}}(\mathbf{x}, \mathbf{k}_c) | c \in [1, C]\} \in \mathbb{R}^{1 \times C}$.

In the association graph, we further build the edges cross the task and class graphs to enable knowledge transfer between them. Formally, we define the edge connecting a class node and a task node as:

$$A_{\mathcal{C} \sim \mathcal{T}}(\mathbf{k}_c, \mathbf{v}_t) = \frac{\exp(-\|(\mathbf{k}_c - \mathbf{v}_t)/\alpha_{\mathcal{P}}\|_2^2/2)}{\sum_{t'=1}^{T} \exp(-\|(\mathbf{k}_c - \mathbf{v}_{t'})/\alpha_{\mathcal{P}}\|_2^2/2)}, \tag{5}$$

where $\alpha_{\mathcal{P}}$ is a fixed scaling factor. This metric function without learnable parameters aims to alleviate the overfitting of the learned connections between the task and its observed classes. The adjacency matrix between the task and class graphs is denoted by $\mathbf{A}_{\mathcal{C} \sim \mathcal{T}} = \{A_{\mathcal{C} \sim \mathcal{T}}(\mathbf{k}_c, \mathbf{v}_t) | c \in [1, C], t \in [1, T]\} \in \mathbb{R}^{C \times T}$. Thus, the whole association graph $\mathcal{G} = (\mathbf{V}, \mathbf{A})$ with three types of nodes can be formulated as:

$$\mathbf{V} = (\mathbf{V}_{\mathcal{T}}; \mathbf{V}_{\mathcal{C}}; \mathbf{V}_{\mathcal{X}}), \mathbf{A} = \begin{bmatrix} \mathbf{A}_{\mathcal{T}} & \mathbf{A}_{\mathcal{C} \sim \mathcal{T}}^{\top} & \mathbf{A}_{\mathcal{X} \sim \mathcal{T}}^{\top} \\ \mathbf{A}_{\mathcal{C} \sim \mathcal{T}} & \mathbf{A}_{\mathcal{C}} & \mathbf{A}_{\mathcal{X} \sim \mathcal{C}}^{\top} \\ \mathbf{A}_{\mathcal{X} \sim \mathcal{T}} & \mathbf{A}_{\mathcal{X} \sim \mathcal{C}} & 1 \end{bmatrix}. \tag{6}$$

**Knowledge transfer by message passing.** With the constructed association graph, we perform message passing in the association graph via a multi-layer Graph Neural Network (GNN). In general, our model can work with various GNN architectures. In this work, we apply GraphSAGE [23]. To simplify, we use $\mathbf{h}_i$ to represent one node in the association graph $\mathcal{G}$, which could be a task, class or instance node. The $l$-th layer of GNNs can be written as:

$$\mathbf{h}_i^{(l)} = \mathbf{U}^l \text{Concat}\Big(\text{Mean}\big(\{\text{ReLU}(\mathbf{W}^l \mathbf{h}_j^{(l-1)}), \mathbf{h}_j \in \mathcal{N}_k(\mathbf{h}_i)\}\big), \mathbf{h}_i^{(l-1)}\Big), \tag{7}$$

where $\mathbf{h}_i^{(l)}$ denotes the node embedding by the $l$-th GNN layer and $\mathbf{U}^l$ and $\mathbf{W}^l$ are learnable weight matrices of the $l$-th GNN layer. $l \in \{1, 2, ..., L\}$ with $L$ denoting the number of GNN layers. $\mathbf{h}_i^{(0)}$ is initialized as the nodes defined above. $\mathcal{N}_k(\mathbf{h}_i)$ denotes the top-$k$ neighbors of the node $\mathbf{h}_i$ according to the adjacency matrix between all nodes $\mathbf{A}$. By message passing, each instance is refined with the categorical information stored in the association graph. As a result, the instance will gain more discriminative and informative representations. The enhanced instance feature is formulated as $\hat{\mathbf{V}}_{\mathcal{X}} = \{\mathcal{E}(\hat{\mathbf{x}})\}$. We compute the prediction for the enhanced feature as $p(\mathbf{y}|\mathbf{x}) = p(\mathbf{y}|\mathcal{E}(\hat{\mathbf{x}}), \mathbf{f}_t)$, where $\mathbf{f}_t$ denotes the corresponding task-specific classifier.

## 3.2 Assignment entropy maximization

Category shifts in multi-task classification yield spurious correlations between tasks and their corresponding observed classes. This means the edges between each task and its observed classes have considerably higher weights than other edges. The knowledge transfer between tasks and classes will be dominated by these spurious correlations, hindering their missing classes from exploiting the categorical information.

To tackle this problem, we propose assignment entropy maximization to encourage each class node to balance the weights of its edges with all tasks. Formally, the assignment entropy for the $c$-th class is formulated as:

$$\mathbf{H}(\mathbf{k}_c) = -\sum_{t=1}^{T} A_{\mathcal{C}\sim\mathcal{T}}(\mathbf{k}_c, \mathbf{v}_t) \log A_{\mathcal{C}\sim\mathcal{T}}(\mathbf{k}_c, \mathbf{v}_t). \tag{8}$$

By maximizing the assignment entropy, weights are balanced for each class, which enables all tasks to fully utilize the categorical information for their missing classes. Intuitively, each class node is task-agnostic when the assignment entropy reaches its maximum values. In this ideal case, the model eliminates the spurious correlations between the class and its corresponding tasks in the graph.

By combining the assignment entropy maximization and cross-entropy minimization of the classifiers, we have the final objective as follows:

$$\mathcal{L} = \frac{1}{T}\sum_{t=1}^{T} \mathcal{L}_{\text{CE}}(\mathcal{D}_t) + \beta \frac{1}{C}\sum_{c=1}^{C} \mathcal{L}_{\text{AE}}(\mathbf{k}_c), \tag{9}$$

where $\mathcal{L}_{\text{CE}} = \mathbb{E}_{\mathcal{D}_t}[-\log p(\mathcal{D}_t|\mathcal{G})]$ and $\mathcal{L}_{\text{AE}}(\mathbf{k}_c) = -\mathbf{H}(\mathbf{k}_c)$. $\beta$ is introduced to balance the importance of the cross-entropy and assignment entropy losses. We provide the training and inference algorithms in the supplemental materials.

## 4 Related Works

**Multi-task learning.** Multi-task learning [9] aims to learn several related tasks simultaneously and improve their overall performance. The task relatedness is learned by many different aspects of the model, e.g., the loss functions [34, 29], gradient space [45, 65], parameter space [37, 4], or representation space [2, 40, 22]. The basic idea of sharing information from multiple tasks has been successfully applied under different settings, including the single-input multi-output setting [45, 65, 48], the multi-input multi-output setting [2, 37, 46], and using meta-learning [16, 54, 1]. In the single-input multi-output setting [24, 15], tasks are defined by different supervision information included in the same input. In meta-learning, tasks are sampled from one task distribution and different tasks have different category spaces [54, 72]. In multi-input multi-output, tasks follow different data distributions since they are collected from different visual modalities or equipment [37, 46, 69, 70]. It remains unexplored to investigate category shifts in multi-task classification, despite being common in various realistic scenarios.

**Category shifts.**    Category shifts denotes that training data collected from different domains may not completely share their categories, which is first proposed by [60] for domain adaptation. [36] and [42] challenge domain generalization with category shifts, where the change of domains is always followed by the change of categories. Category shift is a common scenario in real-world applications since it relaxes the requirement on the shared category set among source domains [60, 10]. As a result, category shift has drawn increasing attention and has been applied to a wide range of learning tasks, including fault diagnosis [14], skin lesion classification [31] and remote sensing image classification [38]. In this paper, we develop a new multi-task learning scenario in which individual tasks do not contain complete training data for the categories in the test set. Unlike domain adaptation and generalization, which only focus on the unidirectional knowledge transfer from source domains to a target domain, our multi-task classification encourages simultaneous bidirectional knowledge transfer between any paired domain to enable efficient predictions for both tasks. To the best of our knowledge, we are the first to address category shifts in multi-task classification.

**Exploring graph structure.**    Several recent works prove that graphs are effective in modeling label correlation [11, 32, 30, 33], task relationships [63, 8, 25], meta-paths [27] and representation learning [61]. For multi-label classification, [32] formulates the multi-label predictions as a conditional graphical lasso inference problem, while [11] utilizes a graph convolutional network to propagate information between multiple labels and consequently learn inter-dependent classifiers for each of the image labels. For multi-task learning, some works learn the relationship between multiple tasks by message passing over a graph neural network [35, 21]. [41, 6] explore the graph structure to produce higher quality node embeddings on the graph-structured data. For few-shot learning, [62] and [63] design the hand-crafted and automatically constructed meta-knowledge graph to provide meta knowledge for each task. Different from these methods, we address the new challenge, category shifts in multi-task classification, which yields a novel graph construction. Particularly, our work shares the high-level goal with [33] in terms of joint inference over multiple heterogeneous graphs with within-graph relations and cross-graph interactions. With respect to the form of information exchange, the main differences are: [33] uses the spectral graph product and label propagation operators to exchange information between heterogeneous nodes, while our work performs message passing (e.g., GNNs). Moreover, [33] infers the unobserved multi-relations with observed multi-relations across the graphs. In contrast, our work refines each instance node with the categorical information from heterogeneous nodes. [27, 61] are related to our work since both of them construct new graph structures to enable knowledge transfer across graphs. [27] multiplies two adjacency matrices (edge types) of the heterogeneous graph to automatically learn new meta-paths. In contrast, our method constructs the connections between heterogeneous nodes by directly computing the similarities between any pair-wise nodes with different metric functions. [61] fixes the edges between heterogeneous nodes to one or zero. By contrast, our method constructs the connections between each heterogeneous node through the similarity between these nodes.

**Heterogeneous GNNs and relational graph models.**    Heterogeneous GNNs models [67, 57, 27, 17, 66, 39] have a similar spirit to our work in dealing with heterogeneous graphs, however the problem settings and technical implementations are fundamentally different. HetGNN-based methods focus on graph data (e.g., academic graph and review graph data) and need pre-processing modules for each node type to encode heterogeneous contents as a fixed-size embedding. Most HetGNN-based methods [67, 57, 27, 17] depend on a hierarchical architecture to aggregate content embeddings of heterogeneous neighbors for each node, which contains node-level (intra-metapath) and semantic-level (inter-metapath) aggregations. Our work constructs an association graph from the non-graph data and enables pair-wise nodes to transfer knowledge whether they are from the same type or not. Thus, our method is suitable for solving the complex knowledge transfer in multi-task classification with category shifts. Moreover, relational graph models [44, 51, 71, 7] are also related to our work since both aim to fully utilize the structure information in the graph. However, the main difference between them is that most relational graph models [44, 51, 71, 7] are designed for homogeneous graphs, while the proposed graph model handles heterogeneous nodes.

**Out-of-distribution generalization.**    These methods are relevant in the sense that both deal with several domains that share the same label space. However, out-of-distribution generalization methods [58, 47, 73, 59, 55, 68, 74, 3, 12, 13] focus on the single-directional knowledge transfer from the source domain(s) to the target domain(s), while our setting aims to learn a bi-directional knowledge transfer between pair-wise domains (tasks). Moreover, due to the various missing classes per task,

the bi-directional knowledge transfer among domains (tasks) varies across different pairs of classes. Thus, the knowledge transfer in our setting is more complex than out-of-distribution generalization.

## 5    Experiments and Results

**Datasets.**    We conduct experiments on three common multi-task classification benchmarks and a skin lesion classification dataset to evaluate the effectiveness of our proposed method.

`Office-Home` [53] contains images from four domains/tasks: Artistic, Clipart, Product and Real-world. Each task contains images from 65 object categories collected under office and home settings. There are about $15,500$ images in total.

`Office-Caltech` [18] contains the ten categories shared between Office-31 [43] and Caltech-256 [19]. One task uses data from Caltech-256, and the other three tasks use data from Office-31, whose images were collected from three distinct domains/tasks, namely Amazon, Webcam and DSLR. There are $8 \sim 151$ samples per category per task, and $2,533$ images in total.

`ImageCLEF` [37], the benchmark for the ImageCLEF domain adaptation challenge, contains 12 common categories shared by four public datasets/tasks: Caltech-256, ImageNet ILSVRC 2012, Pascal VOC 2012, and Bing. There are $2,400$ images in total.

`Skin-Lesion` contains three skin lesion classification tasks: HAM10000 [49], Dermofit [5] and Derm7pt [28]. Tasks are collected from different hospitals or healthcare facilities. In this dataset, each task contains a subset of the following classes: melanocytic nevus, melanoma, basal cell carcinoma, dermatofibroma, benign keratosis and vascular lesion.

**Experimental setup.**    We explore the effect of varying degrees of category shifts in these datasets by different missing rates in the training label spaces. For the three common multi-classification benchmarks, we set the missing rates for the three benchmarks as 75%, 50%, 25%, 0%, denoting that each task cannot access the training data of 75% (or 50%, 25%, 0%) of the categories. For simplicity, we use the same missing rate for all tasks in each setting, which is achieved by assigning the same number of missing classes for each task. Since the union of the training label space of all tasks equals the test label space in the proposed setting, 75% is the largest missing rate for the common multi-task classification datasets that have data from four tasks. With the 75% missing rate, data from each category is only accessible in one task. By contrast, when the missing rate is set to 0%, the problem degrades to the conventional multi-task classification without category shifts, i.e., each task has complete training data from all classes. Since `Skin-Lesion` has three tasks, we set the missing rates as 67%, 33%, 0%. For a fair comparison, the assignment of missing classes for different missing rates and datasets is shared for all methods. We use ResNet-18 [26] as the backbone for all experiments and deploy the graph in the output space of the backbone. We provide the code and the missing class assignment of each dataset in the supplemental materials.

**Metrics.**    The average multi-task classification accuracy (%, top-1) along with $95\%$ confidence intervals from five runs are reported across all tasks. In order to evaluate the model's generalization power from observed classes to missing classes of each task, test instances come from observed classes and missing classes. We report the average accuracy of missing classes and observed classes of all tasks as $A_m$ and $A_o$, respectively. Moreover, we apply the harmonic mean to show the overall performance on both missing and observed classes, which is denoted by $H = \frac{2 \times A_m \times A_o}{A_m + A_o}$.

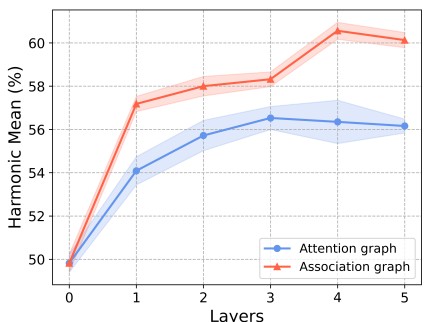

Figure 3: Comparisons between the attention graph and proposed association graph. Our association graph consistently performs better than the attention graph with different message passing layers.

**Benefit of the association graph.**    To show the benefit of the proposed association graph, we conduct experiments with increasing numbers of the message passing layers, where $L=0$ denotes

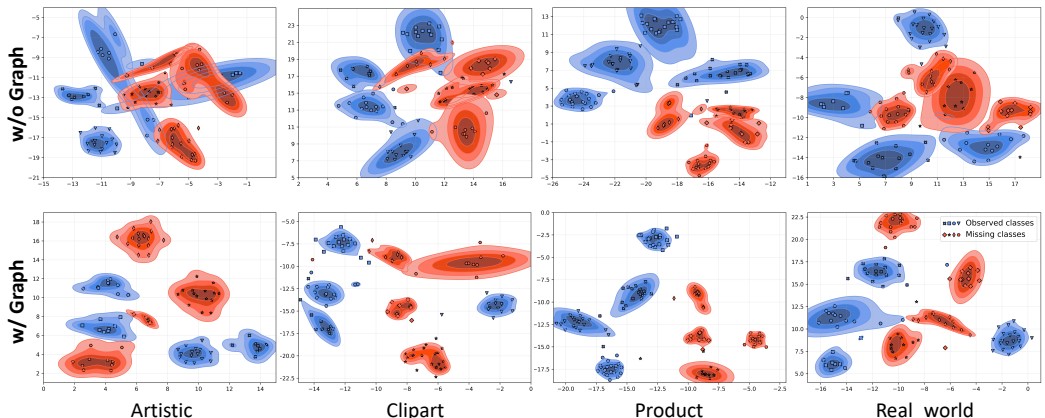

Figure 4: Benefit of the association graph in feature learning. Visualization of features without (upper row) and with the graph (lower row) are shown, where each column corresponds to a task from `Office-Home`. Different shapes denote different classes, while observed and missing classes are in blue and red, respectively. The proposed graph distinguishes each missing class from the observed classes, which tend to collapse together due to the category shifts.

Table 1: Benefit of the assignment entropy maximization in our model on `Office-Home` under the setting of missing 75% classes for each task. The assignment entropy maximization improves the overall performance on both missing and observed classes.

| Method | $A_m$ | $A_o$ | $H$ | Average Assignment Entropy |
|---|---|---|---|---|
| w/o $\mathcal{L}_{\text{AE}}$ | $44.65 \pm 0.29$ | $\mathbf{87.59} \pm 0.35$ | $58.26 \pm 0.31$ | $0.13 \pm 0.05$ |
| w/ $\mathcal{L}_{\text{AE}}$ | $\mathbf{47.51} \pm 0.39$ | $87.16 \pm 0.34$ | $\mathbf{60.59} \pm 0.24$ | $\mathbf{0.84} \pm 0.07$ |

the model without the graph and thus without knowledge transfer among graph nodes, shown in Figure 3. As the number $L$ increases, our model performs increasingly better with the peak at $L=4$, surpassing the model without graphs by a large margin. By stacking multiple layers of GNNs, each node can eventually incorporate the knowledge from the large-hop neighbors across the entire graph to reduce the generalization gap between the training and test sets. The results demonstrate that the task and class-specific knowledge stored in the graph is important to enhance the categorical information in the features, which enables them to be more discriminative. In the following experiments, we set $L=4$ for our model.

To better understand the benefit of the association graph in feature learning, we visualize the distributions of samples from observed and missing classes of each task on `Office-Home` by t-SNE [50]. Figure 4 shows that the association graph reduces the overlap of the distributions of missing classes (shapes in red) and updates the features of each class to be more clustered. We therefore conclude that the association graph enhances the categorical information of the instance features from both observed classes and missing classes, making them more distinguishable.

**Association graph vs. attention graph.** We compare the association graph with the attention graph [52], which incorporates the same self-attention strategy for all nodes. Different from the attention graph, the proposed association graph adopts different learnable metric functions for different types of nodes. In Figure 3, the association graph performs consistently better than the attention graph with different numbers of message passing layers. The association graph with different learnable metric functions is suitable for modeling complex relationships among different types of nodes, leading to improvements over the attention graph.

**Benefit of assignment entropy maximization.** We investigate the benefit of the proposed assignment entropy maximization in our model. As shown in Table 1, the assignment entropy maximization regularization considerably improves the overall performance. This is reasonable since the assignment entropy maximization balances the edge weights between each class and all task nodes. Thus, the spurious correlation between task and class nodes is reduced, which enables each task to fully utilize the categorical information provided by other tasks.

**Importance of knowledge transfer by message passing.** To investigate the importance of knowledge transfer, we conduct experiments on `Office-Home` with different neighbor sizes for each node during message passing, which reflects the amount of the transferred knowledge.

With the size of 0, each node does not utilize the transferred knowledge. With the maximum size (which is 70 in this dataset), each node aggregates the knowledge from all other nodes on the graph. As shown in Table 2, we find that the best performance happens with the largest neighbor size, which indicates the importance of passing messages throughout the whole graph for knowledge transfer.

Table 2: Performance with different neighbor sizes on `Office-Home`. The largest size performs best.

| $|\mathcal{N}_k|$ | $A_m$ | $A_o$ | $H$ |
|---|---|---|---|
| 0 | $36.84 \pm 0.31$ | $83.09 \pm 0.93$ | $49.82 \pm 0.76$ |
| 8 | $45.48 \pm 0.21$ | $84.82 \pm 0.67$ | $58.45 \pm 0.34$ |
| 16 | $46.14 \pm 0.34$ | $84.07 \pm 0.55$ | $58.83 \pm 0.28$ |
| 32 | $46.10 \pm 0.25$ | $84.89 \pm 0.42$ | $58.80 \pm 0.35$ |
| 64 | $\mathbf{47.51} \pm 0.37$ | $85.95 \pm 0.38$ | $60.20 \pm 0.22$ |
| 70 (max.) | $\mathbf{47.51} \pm 0.39$ | $\mathbf{87.16} \pm 0.34$ | $\mathbf{60.59} \pm 0.24$ |

Table 3: Comparative results with different missing rates on `Office-Home`, `Office-Caltech` and `ImageCLEF`. Our method is a consistent top performer on missing and observed classes.

| Method | Missing Rate ($\gamma$) | Office-Home | | | Office-Caltech | | | ImageCLEF | | |
|---|---|---|---|---|---|---|---|---|---|---|
| | | $A_m$ | $A_o$ | $H$ | $A_m$ | $A_o$ | $H$ | $A_m$ | $A_o$ | $H$ |
| STL | | 0.00 | **88.25** | 0.00 | 0.00 | **98.53** | 0.00 | 0.00 | **95.00** | 0.00 |
| ERM [20] | | 36.45 | 83.53 | 49.32 | 47.43 | 97.28 | 62.98 | 71.94 | 80.00 | 75.55 |
| PCGrad [65] | 75% | 36.99 | 83.30 | 49.56 | 49.84 | 96.43 | 64.93 | 71.94 | 83.33 | 76.92 |
| WeighLosses [29] | | 37.39 | 82.92 | 50.26 | 49.39 | 96.43 | 64.46 | 72.22 | 80.83 | 76.08 |
| **Ours** | | **47.51** | 87.16 | **60.59** | **55.47** | 98.12 | **70.55** | **75.28** | 85.00 | **79.45** |
| STL | | 0.00 | 84.37 | 0.00 | 0.00 | **98.61** | 0.00 | 0.00 | **88.33** | 0.00 |
| ERM [20] | | 50.96 | 81.89 | 62.14 | 77.33 | 97.43 | 85.09 | 76.67 | 84.58 | 80.36 |
| PCGrad [65] | 50% | 50.95 | 82.52 | 62.39 | 80.29 | 97.43 | 87.28 | 74.58 | 82.92 | 78.46 |
| WeighLosses [29] | | 51.65 | 82.38 | 62.84 | 76.54 | 97.27 | 84.60 | 75.42 | 85.42 | 79.94 |
| **Ours** | | **54.65** | 83.57 | **65.54** | **88.65** | 98.15 | **92.83** | **78.33** | 87.08 | **82.20** |
| STL | | 0.00 | 82.06 | 0.00 | 0.00 | 98.07 | 0.00 | 0.00 | 83.06 | 0.00 |
| ERM [20] | | 54.09 | 81.34 | 64.51 | 94.27 | 97.42 | 95.76 | 74.17 | 85.00 | 78.90 |
| PCGrad [65] | 25% | 52.43 | 80.81 | 63.18 | 92.96 | 97.92 | 95.20 | 76.67 | 82.22 | 79.12 |
| WeighLosses [29] | | 53.60 | 81.38 | 64.03 | 93.84 | 97.81 | 95.69 | 76.67 | 83.89 | 79.88 |
| **Ours** | | **56.74** | **82.94** | **67.12** | **97.35** | **98.51** | **97.92** | **80.00** | **85.28** | **82.48** |
| STL | | - | 79.29 | - | - | 98.13 | - | - | 81.67 | - |
| ERM [20] | | - | 80.99 | - | - | 98.22 | - | - | 84.79 | - |
| PCGrad [65] | 0% | - | 81.41 | - | - | 98.02 | - | - | 82.71 | - |
| WeighLosses [29] | | - | 81.78 | - | - | 98.24 | - | - | 82.75 | - |
| **Ours** | | - | **82.01** | - | - | **98.26** | - | - | **86.04** | - |

**Effect of different degrees of category shifts.** We evaluate the proposed method on the three benchmarks with different missing rates in Table 3. ERM [20] is a simple baseline that mixes all training data together and trains the shared model. Our method achieves the best overall performance on all three benchmarks under each missing rate in terms of the harmonic mean of the accuracy of observed and missing classes. We note that 75% is the most severe category shifts in multi-task classification, which demonstrates there are no overlap categories between tasks. On `Office-Home` with the 75% missing rate, our model surpasses the second best method, i.e., WeighLosses [29], by a large margin of 10.33%, in terms of the harmonic mean. The consistent improvements on all benchmarks with different missing rates demonstrate that the association graph is effective in addressing the category shifts for multi-task classification. More detailed results with 95% confidence intervals are provided in the supplemental materials. Moreover, we also provide the results on the realistic medical dataset, i.e., `Skin-Lesion` with missing rates of 67%, 33% and 0% in Table 4. Our model achieves the best overall performance, which again confirms the effectiveness of our method.

It is worth mentioning that with $\gamma=0\%$ the setting reduces to traditional multi-task learning without category shifts, where all classes are observed by each task during training. In this case, the results of missing classes $A_m$ and the harmonic mean $H$ are not available. Our method still outperforms other baselines in all datasets. We conclude that our association graph better utilizes the shared knowledge to improve overall performance for all tasks, which also holds for settings without category shifts.

Table 4: Comparative results with different missing rates on the medical dataset `Skin-Lesion`. Our method achieves the best overall performance on both missing and observed classes.

| Method | $\gamma = 67\%$ | | | $\gamma = 33\%$ | | | $\gamma = 0\%$ | | |
|---|---|---|---|---|---|---|---|---|---|
| | $A_m$ | $A_o$ | $H$ | $A_m$ | $A_o$ | $H$ | $A_m$ | $A_o$ | $H$ |
| STL | 0.00 | **97.99** | 0.00 | 0.00 | **87.32** | 0.00 | - | 84.33 | - |
| ERM [20] | 8.74 | 93.95 | 15.16 | 15.52 | 84.24 | 25.96 | - | 83.48 | - |
| PCGrad [65] | 8.04 | 91.62 | 14.51 | 14.28 | 82.77 | 23.53 | - | 84.11 | - |
| WeighLosses [29] | 7.73 | 89.68 | 13.07 | 14.25 | 85.56 | 24.35 | - | 84.20 | - |
| **Ours** | **10.82** | 90.29 | **18.17** | **16.58** | 86.62 | **27.21** | - | **85.98** | - |

## 6 Conclusion

We address category shifts in multi-task classification, which is challenging yet more realistic since individual tasks often do not contain complete training data for the categories in the test set. To tackle this, we propose to learn an association graph to transfer knowledge among tasks for missing classes, which enhances the categorical information of each instance, making them more discriminative. To avoid the spurious correlations between task and class nodes, we introduce assignment entropy maximization, enabling all tasks to fully utilize the categorical information for missing classes. To the best of our knowledge, we are the first to address the challenge in multi-task classification. We conduct ablation studies to demonstrate the effectiveness of the proposed association graph and assignment entropy maximization in our model. The superior performance on three multi-task classification benchmarks and the medical dataset for skin lesion classification further substantiates the benefits of our proposal.

## Acknowledgment

This work is financially supported by the Inception Institute of Artificial Intelligence, the University of Amsterdam and the allowance Top consortia for Knowledge and Innovation (TKIs) from the Netherlands Ministry of Economic Affairs and Climate Policy.

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
