# A  Algorithm

---

**Algorithm 1** Association Graph Learning (TRAINING TIME)

---

**Require:** $\{\mathcal{D}_t^{tr}\}_{t=1}^T$: Training sets of all tasks; $T$: Number of tasks; $C$: Number of all classes; $\mathcal{E}$: Shared feature extractor; $\mathbf{W}_{\mathcal{T}}, \mathbf{W}_{\mathcal{C}}$: Parameters of metric functions in the association graph; $L$: Number of GNN layers; $\{\mathbf{W}^l\}_{l=1}^L$: Parameters of all GNN layers; $\{\mathbf{f}_t\}_{t=1}^T$: Task-specific classifiers; $\lambda$: Learning rate.
 1: Initialize the feature extractor $\mathcal{E}$.
 2: Initialize all task and class nodes $\{\mathbf{v}_t\}_{t=1}^T, \{\mathbf{k}_c\}_{c=1}^C$.
 3: **while** not done **do**
 4:     Sample a batch of each task, and embed each instance into the feature space $\mathcal{E}(\mathbf{x})$.
 5:     **for** $t = 1$ to $T$ **do**
 6:         Collect instances from the $t$-th task in the mini-batch $\{\mathbf{x}_i\}_{i=1}^{N_t^{tr}}$.
 7:         Aggregate the task representation in the mini-batch $\mathbf{v}_{t(mini-batch)} \leftarrow \frac{1}{N_t^{tr}} \sum_{i=1}^{N_t^{tr}} \mathcal{E}(\mathbf{x}_i)$.
 8:         Compute the task node by moving average $\mathbf{v}_t \leftarrow 0.9 * \mathbf{v}_t + 0.1 * \mathbf{v}_{t(mini-batch)}$.
 9:     **end for**
10:     Construct the task graph $\mathcal{G}_{\mathcal{T}}$ with edge weights in equation (2).
11:     **for** $c = 1$ to $C$ **do**
12:         Collect instances from the $c$-th class in the mini-batch $\{\mathbf{x}_m\}_{m=1}^{N_c^{tr}}$.
13:         Aggregate the class information in the mini-batch $\mathbf{k}_{c(mini-batch)} \leftarrow \frac{1}{N_c^{tr}} \sum_{m=1}^{N_c^{tr}} \mathcal{E}(\mathbf{x}_m)$.
14:         Compute the class node by moving average $\mathbf{k}_c \leftarrow 0.9 * \mathbf{k}_c + 0.1 * \mathbf{k}_{c(mini-batch)}$.
15:     **end for**
16:     Construct the class graph $\mathcal{G}_{\mathcal{C}}$ with edge weights in equation (4).
17:     Compute the edges between all three different types of nodes and construct the association graph in equation (6).
18:     **for** $l = 1$ to $L$ **do**
19:         Apply GNN with the parameter $\mathbf{W}^l$ on the association graph $\mathcal{G}$ and update all nodes in the graph.
20:     **end for**
21:     Compute the assignment entropy for each class node in equation (8).
22:     Update parameters by $\mathcal{E}, \mathbf{W}_{\mathcal{T}}, \mathbf{W}_{\mathcal{C}}, \{\mathbf{W}^l\}_{l=1}^L, \{\mathbf{f}_t\}_{t=1}^T - \lambda \nabla_{\mathcal{E}, \mathbf{W}_{\mathcal{T}}, \mathbf{W}_{\mathcal{C}}, \{\mathbf{W}^l\}_{l=1}^L, \{\mathbf{f}_t\}_{t=1}^T} \mathcal{L}$.
23: **end while**

---

In this paper, we propose to learn an association graph to address category shifts in multi-task classification. For clarity, we provide the algorithms during training and test in Algorithm 1 and Algorithm 2, respectively.

---

**Algorithm 2** Association Graph Learning (TEST TIME)

---

**Require:** $\mathbf{x}_t$: one test instance from the $t$-th task; $\mathcal{E}$: Trained the feature extractor; $\mathcal{G}_{\mathcal{T}}, \mathcal{G}_{\mathcal{C}}$: Trained task and class graph; $L$: Number of GNN layers; $\{\mathbf{W}^l\}_{l=1}^L$: Trained parameters of all GNN layers; $\mathbf{f}_t$: The trained task-specific classifier.
 1: Embed each test instance for the $t$-th task into the feature space $\mathcal{E}(\mathbf{x}_t)$.
 2: Construct the association graph $\mathcal{G}$ to connect the trained task $\mathcal{G}_{\mathcal{T}}$ and class graph $\mathcal{G}_{\mathcal{C}}$ and the test instance node.
 3: **for** $l = 1$ to $L$ **do**
 4:     Apply GNN with the parameter $\mathbf{W}^l$ on the association graph $\mathcal{G}$ and update the instance node in the graph.
 5: **end for**
 6: Obtain the updated instance node $\mathcal{E}(\hat{\mathbf{x}}_t)$.
 7: Predict the test instance with the task-specific classifier $p(\hat{\mathbf{y}}_t | \mathbf{f}_t, \mathcal{E}(\hat{\mathbf{x}}_t))$.

---

Table B.1: The observed classes of each task on `Office-Home` with different missing rates. $\gamma = 0\%$ denotes all task share the entire label space.

| Missing rates | Artistic | Clipart | Product | Real_world |
|---|---|---|---|---|
| $\gamma = 75\%$ | 'Alarm_Clock', 'Bottle', 'Fan', 'Flowers', 'Fork', 'Glasses', 'Helmet', 'Kettle', 'Knives', 'Lamp_Shade', 'Push_Pin', 'Radio', 'Refrigerator', 'Shelf', 'Soda', 'Spoon' | 'Batteries', 'Computer', 'Drill', 'Folder', 'Hammer', 'Keyboard', 'Marker', 'Monitor', 'Mug', 'Pan', 'Pen', 'Pencil', 'Ruler', 'Screwdriver', 'TV', 'Table', 'Toys' | 'Calculator', 'Calendar', 'Chair', 'Couch', 'Desk_Lamp', 'Flipflops', 'Laptop', 'Mop', 'Mouse', 'Notebook', 'Printer', 'Scissors', 'Sneakers', 'Speaker', 'Trash_Can', 'Webcam' | 'Backpack', 'Bed', 'Bike', 'Bucket', 'Candles', 'Clipboards', 'Curtains', 'Eraser', 'Exit_Sign', 'File_Cabinet', 'Oven', 'Paper_Clip', 'Postit_Notes', 'Sink', 'Telephone', 'ToothBrush' |
| $\gamma = 50\%$ | 'Alarm_Clock', 'Batteries', 'Bike', 'Bottle', 'Bucket', 'Candles', 'Desk_Lamp', 'Fan', 'File_Cabinet', 'Flowers', 'Fork', 'Glasses', 'Hammer', 'Helmet', 'Kettle', 'Knives', 'Lamp_Shade', 'Laptop', 'Marker', 'Mop', 'Mug', 'Paper_Clip', 'Pencil', 'Push_Pin', 'Radio', 'Refrigerator', 'Screwdriver', 'Shelf', 'Sink', 'Soda', 'Spoon', 'ToothBrush' | 'Batteries', 'Bed', 'Bottle', 'Calendar', 'Computer', 'Drill', 'Fan', 'Flowers', 'Folder', 'Fork', 'Hammer', 'Keyboard', 'Marker', 'Monitor', 'Mouse', 'Mug', 'Notebook', 'Pan', 'Pen', 'Pencil', 'Postit_Notes', 'Printer', 'Push_Pin', 'Ruler', 'Scissors', 'Screwdriver', 'Soda', 'Spoon', 'TV', 'Table', 'Telephone', 'Toys' | 'Backpack', 'Calculator', 'Calendar', 'Chair', 'Clipboards', 'Computer', 'Couch', 'Curtains', 'Desk_Lamp', 'Drill', 'Exit_Sign', 'File_Cabinet', 'Flipflops', 'Folder', 'Glasses', 'Helmet', 'Kettle', 'Keyboard', 'Laptop', 'Monitor', 'Mop', 'Mouse', 'Notebook', 'Oven', 'Pan', 'Printer', 'Scissors', 'Sneakers', 'Speaker', 'TV', 'Trash_Can', 'Webcam' | 'Alarm_Clock', 'Backpack', 'Bed', 'Bike', 'Bucket', 'Calculator', 'Candles', 'Chair', 'Clipboards', 'Couch', 'Curtains', 'Eraser', 'Exit_Sign', 'Flipflops', 'Knives', 'Lamp_Shade', 'Oven', 'Paper_Clip', 'Pen', 'Postit_Notes', 'Radio', 'Refrigerator', 'Shelf', 'Sink', 'Sneakers', 'Speaker', 'Table', 'Telephone', 'ToothBrush', 'Toys', 'Trash_Can', 'Webcam' |
| $\gamma = 25\%$ | 'Batteries', 'Bed', 'Bike', 'Bottle', 'Bucket', 'Calendar', 'Candles', 'Chair', 'Computer', 'Drill', 'Eraser', 'Fan', 'Flowers', 'Folder', 'Fork', 'Glasses', 'Hammer', 'Helmet', 'Keyboard', 'Knives', 'Laptop', 'Marker', 'Monitor', 'Mop', 'Mouse', 'Mug', 'Notebook', 'Pan', 'Paper_Clip', 'Pen', 'Pencil', 'Postit_Notes', 'Printer', 'Push_Pin', 'Radio', 'Ruler', 'Scissors', 'Screwdriver', 'Sink', 'Soda', 'Speaker', 'Spoon', 'TV', 'Table', 'Telephone', 'Toys', 'Trash_Can', 'Webcam' | 'Batteries', 'Bed', 'Bike', 'Bottle', 'Bucket', 'Calendar', 'Candles', 'Chair', 'Computer', 'Drill', 'Eraser', 'Fan', 'Flowers', 'Folder', 'Fork', 'Glasses', 'Hammer', 'Helmet', 'Keyboard', 'Knives', 'Laptop', 'Marker', 'Monitor', 'Mop', 'Mouse', 'Mug', 'Notebook', 'Pan', 'Paper_Clip', 'Pen', 'Pencil', 'Postit_Notes', 'Printer', 'Push_Pin', 'Radio', 'Ruler', 'Scissors', 'Screwdriver', 'Sink', 'Soda', 'Speaker', 'Spoon', 'TV', 'Table', 'Telephone', 'Toys', 'Trash_Can', 'Webcam' | 'Alarm_Clock', 'Backpack', 'Batteries', 'Calculator', 'Calendar', 'Chair', 'Clipboards', 'Computer', 'Couch', 'Curtains', 'Desk_Lamp', 'Drill', 'Exit_Sign', 'Fan', 'File_Cabinet', 'Flipflops', 'Flowers', 'Folder', 'Fork', 'Glasses', 'Hammer', 'Helmet', 'Kettle', 'Keyboard', 'Lamp_Shade', 'Laptop', 'Marker', 'Monitor', 'Mop', 'Mouse', 'Notebook', 'Oven', 'Pan', 'Paper_Clip', 'Pen', 'Printer', 'Refrigerator', 'Ruler', 'Scissors', 'Shelf', 'Sneakers', 'Speaker', 'Spoon', 'TV', 'Table', 'ToothBrush', 'Trash_Can', 'Webcam' | 'Alarm_Clock', 'Backpack', 'Bed', 'Bike', 'Bottle', 'Bucket', 'Calculator', 'Calendar', 'Candles', 'Chair', 'Clipboards', 'Couch', 'Curtains', 'Desk_Lamp', 'Drill', 'Eraser', 'Exit_Sign', 'File_Cabinet', 'Flipflops', 'Kettle', 'Keyboard', 'Knives', 'Lamp_Shade', 'Mouse', 'Mug', 'Notebook', 'Oven', 'Paper_Clip', 'Pen', 'Pencil', 'Postit_Notes', 'Printer', 'Push_Pin', 'Radio', 'Refrigerator', 'Scissors', 'Screwdriver', 'Shelf', 'Sink', 'Sneakers', 'Soda', 'Speaker', 'Table', 'Telephone', 'ToothBrush', 'Toys', 'Trash_Can', 'Webcam' |
| $\gamma = 0\%$ | 'Alarm_Clock', 'Backpack', 'Batteries', 'Bed', 'Bike', 'Bottle', 'Bucket', 'Calculator', 'Calendar', 'Candles', 'Chair', 'Clipboards', 'Computer', 'Couch', 'Curtains', 'Desk_Lamp', 'Drill', 'Eraser', 'Exit_Sign', 'Fan', 'File_Cabinet', 'Flipflops', 'Flowers', 'Folder', 'Fork', 'Glasses', 'Hammer', 'Helmet', 'Kettle', 'Keyboard', 'Knives', 'Lamp_Shade', 'Laptop', 'Marker', 'Monitor', 'Mop', 'Mouse', 'Mug', 'Notebook', 'Oven', 'Pan', 'Paper_Clip', 'Pen', 'Pencil', 'Postit_Notes', 'Printer', 'Push_Pin', 'Radio', 'Refrigerator', 'Ruler', 'Scissors', 'Screwdriver', 'Shelf', 'Sink', 'Sneakers', 'Soda', 'Speaker', 'Spoon', 'TV', 'Table', 'Telephone', 'ToothBrush', 'Toys', 'Trash_Can', 'Webcam' | | | |

# B  Class assignment of all datasets

In this section, we provide the class assignment of all datasets under different missing rates. Table B.1, B.2, B.3 shows the class assignment for `Office-Home`, `Office-Caltech` and `ImageCLEF`, respectively. For the `Skin-Lesion`, each task contains a subset of the following classes: melanocytic nevus (nv), melanoma (mel), basal cell carcinoma (bcc), dermatofibroma (df), benign keratosis (bkl) and vascular lesion (vasc). The class assignment is provided in Table B.4.

The proposed setting is a new multi-task learning scenario. Its practical applications could not be limited by the mentioned assumption in the testing space. In our setting, the test label spaces of different tasks are not forced to be the same since the model predicts instances from different classes and tasks independently during inference. The real significance of this setting is coordinating multiple related tasks to make up for the missing information of each task from other tasks. In order to evaluate the methods under different degrees of category shifts, we set various missing rates of the training set.

Table B.2: The observed classes of each task on `Office-Caltech` with different missing rates. $\gamma = 0\%$ denotes all tasks share the entire label space.

| Missing rates | Amazon | Webcam | DSLR | Caltech |
|---|---|---|---|---|
| $\gamma = 75\%$ | 'keyboard', 'laptop_computer' | 'calculator', 'monitor', 'mouse' | 'bike', 'projector' | 'back_pack', 'headphones', 'mug' |
| $\gamma = 50\%$ | 'headphones', 'keyboard', 'laptop_computer', 'mouse', 'mug' | 'back_pack', 'calculator', 'monitor', 'mouse', 'projector' | 'bike', 'keyboard', 'laptop_computer', 'monitor', 'projector' | 'back_pack', 'bike', 'calculator', 'headphones', 'mug' |
| $\gamma = 25\%$ | 'calculator', 'headphones', 'keyboard', 'laptop_computer', 'mouse', 'mug', 'projector' | 'back_pack', 'calculator', 'keyboard', 'monitor', 'mouse', 'mug', 'projector' | 'back_pack', 'bike', 'calculator', 'headphones', 'laptop_computer', 'monitor', 'projector' | back_pack', 'bike', 'headphones', 'laptop_computer', 'monitor', 'mouse', 'mug' |
| $\gamma = 0\%$ | 'back_pack', 'bike', 'calculator', 'headphones', 'keyboard', 'laptop_computer', 'monitor', 'mouse', 'mug', 'projector' | | | |

Table B.3: The observed classes of each task on `ImageCLEF` with different missing rates. $\gamma = 0\%$ denotes all tasks share the entire label space.

| Missing rates | Caltech | ImageNet | Pascal | Bing |
|---|---|---|---|---|
| $\gamma = 75\%$ | 'bikes', 'computer-monitor', 'school-bus' | 'car-side', 'hummingbird', 'motorbikes' | 'dog', 'people', 'speed-boat' | 'airplanes', 'horse', 'wine-bottle' |
| $\gamma = 50\%$ | 'bikes', 'computer-monitor', 'dog', 'people', 'school-bus', 'speed-boat' | 'bikes', 'computer-monitor', 'dog', 'people', 'school-bus', 'speed-boat' | 'airplanes', 'car-side', 'horse', 'hummingbird', 'motorbikes', 'wine-bottle' | 'airplanes', 'car-side', 'horse', 'hummingbird', 'motorbikes', 'wine-bottle' |
| $\gamma = 25\%$ | 'bikes', 'car-side', 'computer-monitor', 'dog', 'hummingbird', 'motorbikes', 'people', 'school-bus', 'speed-boat' | 'airplanes', 'bikes', 'car-side', 'computer-monitor', 'horse', 'hummingbird', 'motorbikes', 'school-bus', 'wine-bottle' | 'airplanes', 'bikes', 'computer-monitor', 'dog', 'horse', 'people', 'school-bus', 'speed-boat', 'wine-bottle' | 'airplanes', 'car-side', 'dog', 'horse', 'hummingbird', 'motorbikes', 'people', 'speed-boat', 'wine-bottle' |
| $\gamma = 0\%$ | 'airplanes', 'bikes', 'car-side', 'computer-monitor', 'dog', 'horse', 'hummingbird', 'motorbikes', 'people', 'school-bus', 'speed-boat', 'wine-bottle' | | | |

Table B.4: The observed classes of each task on `Skin-Lesion` with different missing rates. $\gamma = 0\%$ denotes all tasks share the entire label space.

| Missing rates | HAM10000 | Dermofit | Derm7pt |
|---|---|---|---|
| $\gamma = 67\%$ | 'bcc', 'nv' | 'mel', 'vasc' | 'bkl', 'df' |
| $\gamma = 33\%$ | 'bkl', 'mel', 'nv', 'vasc' | 'bcc', 'df', 'mel', 'nv' | 'bcc', 'bkl', 'df', 'vasc' |
| $\gamma = 0\%$ | 'bcc', 'bkl', 'df', 'mel', 'nv', 'vasc' | | |

## C  Datasets

`Office-Home` [9] contains images from four domains/tasks: Artistic, Clipart, Product and Real-world. Each task contains images from 65 object categories collected under office and home settings. There are about $15,500$ images in total.

`Office-Caltech` [3] contains the ten categories shared between Office-31 [7] and Caltech-256 [4]. One task uses data from Caltech-256, and the other three tasks use data from Office-31, whose images were collected from three distinct domains/tasks, namely Amazon, Webcam and DSLR. There are $8 \sim 151$ samples per category per task, and $2,533$ images in total.

`ImageCLEF` [6], the benchmark for the ImageCLEF domain adaptation challenge, contains 12 common categories shared by four public datasets/tasks: Caltech-256, ImageNet ILSVRC 2012, Pascal VOC 2012, and Bing. There are $2,400$ images in total.

`Skin-Lesion` contains three skin lesion classification tasks: HAM10000 [8], Dermofit [2] and Derm7pt [5]. Tasks are collected from different hospitals or healthcare facilities. In this dataset, each task contains a subset of the following classes: melanocytic nevus, melanoma, basal cell carcinoma, dermatofibroma, benign keratosis and vascular lesion.

## D  Detailed Results

We provide detailed information for Figure 3 of the paper in Table D.5. The $95\%$ confidence intervals of Table 4 and Table 5 of the paper are shown in Table D.6 and Table D.7. STL is the typical baseline of single-task learning, which learns each task independently.

Table D.5: Benefit of the proposed association graph on `Office-Home` under the setting of missing 75% classes. $L$ denotes the number of massage passing layers. Our association graph consistently performs better than the attention graph with different message passing layers.

| Method | L=0 | L=1 | L=2 | L=3 | L=4 | L=5 |
|---|---|---|---|---|---|---|
| Attention Graph | $49.82_{\pm 0.42}$ | $54.09_{\pm 0.65}$ | $55.72_{\pm 0.70}$ | $56.53_{\pm 0.53}$ | $56.35_{\pm 1.00}$ | $56.16_{\pm 0.32}$ |
| **Association Graph** | $49.82_{\pm 0.42}$ | $\mathbf{57.18}_{\pm 0.35}$ | $\mathbf{58.00}_{\pm 0.45}$ | $\mathbf{58.32}_{\pm 0.34}$ | $\mathbf{60.56}_{\pm 0.39}$ | $\mathbf{60.13}_{\pm 0.35}$ |

## E  More Visualizations

To further show the benefit of the association graph in feature learning, we visualize the test samples from four classes of all tasks in Figure E.1. The model is trained on `Office-Home` with the $75\%$ missing rate. The four classes are observed by `Artistic` during training and missed by other tasks. The visualizations show that the association graph encourages features from the same class of different tasks to be more clustered. This demonstrates that the association graph effectively generalizes the categorical information from seen classes (of `Artistic`) to missing classes (of other tasks), making them more distinguishable.

To show the relationships between tasks and classes, we visualize the similarity matrices between task and class nodes with different numbers of GNN layers. As shown in Figure E.2, the model with the association graph ($L > 0$) obtains more evenly distributed similarities than the model without the graph ($L = 0$). Moreover, when $L = 4$, fewer categories have absolutely dominated tasks (yellow square), which improves the knowledge transferring from observed classes to missing classes for each task.

Table D.6: Comparative results under the proposed setting with different missing rates on `Office-Home`, `Office-Caltech` and `ImageCLEF` using a ResNet-18 backbone. The best performance is in bold. Our method improves the overall performance of both seen and unseen classes.

| Method | Missing Rate | Office-Home $A_u$ | $A_s$ | $H$ | Office-Caltech $A_u$ | $A_s$ | $H$ | ImageCLEF $A_u$ | $A_s$ | $H$ |
|---|---|---|---|---|---|---|---|---|---|---|
| STL | | $0.00 \pm 0.00$ | $\mathbf{88.25} \pm 0.51$ | $0.00 \pm 0.00$ | $0.00 \pm 0.00$ | $\mathbf{98.53} \pm 0.77$ | $0.00 \pm 0.00$ | $0.00 \pm 0.00$ | $\mathbf{95.00} \pm 0.56$ | $0.00 \pm 0.00$ |
| ERM | | $36.45 \pm 0.33$ | $83.53 \pm 0.42$ | $49.32 \pm 0.36$ | $47.43 \pm 0.56$ | $97.28 \pm 0.73$ | $62.98 \pm 0.58$ | $71.94 \pm 0.23$ | $80.00 \pm 0.58$ | $75.55 \pm 0.38$ |
| PCGrad | 75% | $36.99 \pm 0.51$ | $83.30 \pm 0.53$ | $49.56 \pm 0.37$ | $49.84 \pm 0.71$ | $96.43 \pm 0.67$ | $64.93 \pm 0.89$ | $71.94 \pm 0.12$ | $83.33 \pm 0.87$ | $76.92 \pm 0.51$ |
| WeighLosses | | $37.39 \pm 0.35$ | $82.92 \pm 0.50$ | $50.26 \pm 0.34$ | $49.39 \pm 0.92$ | $96.43 \pm 0.33$ | $64.46 \pm 0.52$ | $72.22 \pm 0.54$ | $80.83 \pm 0.86$ | $76.08 \pm 0.76$ |
| **Ours** | | $\mathbf{47.51} \pm 0.32$ | $87.16 \pm 0.44$ | $\mathbf{60.59} \pm 0.35$ | $\mathbf{55.47} \pm 0.20$ | $98.12 \pm 0.49$ | $\mathbf{70.55} \pm 0.28$ | $\mathbf{75.28} \pm 0.37$ | $85.00 \pm 0.52$ | $\mathbf{79.45} \pm 0.36$ |
| STL | | $0.00 \pm 0.00$ | $84.37 \pm 0.29$ | $0.00 \pm 0.00$ | $0.00 \pm 0.00$ | $\mathbf{98.61} \pm 0.41$ | $0.00 \pm 0.00$ | $0.00 \pm 0.00$ | $\mathbf{88.33} \pm 0.27$ | $0.00 \pm 0.00$ |
| ERM | | $50.96 \pm 0.22$ | $81.89 \pm 0.32$ | $62.14 \pm 0.26$ | $77.33 \pm 0.17$ | $97.43 \pm 0.50$ | $85.09 \pm 0.21$ | $76.67 \pm 0.34$ | $84.58 \pm 0.28$ | $80.36 \pm 0.27$ |
| PCGrad | 50% | $50.95 \pm 0.18$ | $82.52 \pm 0.66$ | $62.39 \pm 0.57$ | $80.29 \pm 0.31$ | $97.43 \pm 0.79$ | $87.28 \pm 0.66$ | $75.42 \pm 0.29$ | $82.92 \pm 0.55$ | $78.46 \pm 0.39$ |
| WeighLosses | | $51.65 \pm 0.54$ | $82.38 \pm 0.39$ | $62.84 \pm 0.44$ | $76.54 \pm 0.24$ | $97.27 \pm 0.51$ | $84.60 \pm 0.32$ | $75.42 \pm 0.29$ | $85.42 \pm 0.37$ | $79.94 \pm 0.30$ |
| **Ours** | | $\mathbf{54.65} \pm 0.23$ | $83.57 \pm 0.35$ | $\mathbf{65.54} \pm 0.32$ | $\mathbf{88.65} \pm 0.19$ | $98.15 \pm 0.40$ | $\mathbf{92.83} \pm 0.31$ | $\mathbf{78.33} \pm 0.24$ | $87.08 \pm 0.38$ | $\mathbf{82.20} \pm 0.26$ |
| STL | | $0.00 \pm 0.00$ | $82.06 \pm 0.82$ | $0.00 \pm 0.00$ | $0.00 \pm 0.00$ | $98.07 \pm 0.63$ | $0.00 \pm 0.00$ | $0.00 \pm 0.00$ | $83.06 \pm 0.92$ | $0.00 \pm 0.00$ |
| ERM | | $54.09 \pm 0.36$ | $81.34 \pm 0.47$ | $64.51 \pm 0.42$ | $94.27 \pm 0.86$ | $97.42 \pm 0.39$ | $95.76 \pm 0.21$ | $74.17 \pm 0.44$ | $85.00 \pm 0.73$ | $78.90 \pm 0.36$ |
| PCGrad | 25% | $52.43 \pm 0.52$ | $80.81 \pm 0.32$ | $63.18 \pm 0.33$ | $92.96 \pm 0.49$ | $97.92 \pm 0.51$ | $95.20 \pm 0.36$ | $76.67 \pm 0.51$ | $82.22 \pm 0.73$ | $79.12 \pm 0.46$ |
| WeighLosses | | $53.60 \pm 0.71$ | $81.38 \pm 0.93$ | $64.03 \pm 0.42$ | $93.84 \pm 0.37$ | $97.81 \pm 0.98$ | $95.69 \pm 0.73$ | $76.67 \pm 0.36$ | $83.89 \pm 0.80$ | $79.88 \pm 0.29$ |
| **Ours** | | $\mathbf{56.74} \pm 0.23$ | $82.94 \pm 0.37$ | $\mathbf{67.12} \pm 0.28$ | $\mathbf{97.35} \pm 0.16$ | $\mathbf{98.51} \pm 0.73$ | $\mathbf{97.92} \pm 0.51$ | $\mathbf{80.00} \pm 0.69$ | $85.28 \pm 0.78$ | $\mathbf{82.48} \pm 0.31$ |
| STL | | - | $79.29 \pm 0.35$ | - | - | $98.13 \pm 0.27$ | - | - | $81.67 \pm 0.53$ | - |
| ERM | | - | $80.99 \pm 0.89$ | - | - | $98.22 \pm 0.62$ | - | - | $84.79 \pm 0.57$ | - |
| PCGrad | 0% | - | $81.41 \pm 0.49$ | - | - | $98.02 \pm 0.48$ | - | - | $82.71 \pm 0.39$ | - |
| WeighLosses | | - | $81.78 \pm 0.45$ | - | - | $98.24 \pm 0.56$ | - | - | $82.75 \pm 0.49$ | - |
| **Ours** | | - | $\mathbf{82.01} \pm 0.26$ | - | - | $98.26 \pm 0.39$ | - | - | $\mathbf{86.04} \pm 0.44$ | - |

Table D.7: Comparative results with different missing rates on the medical dataset `Skin-Lesion`. Our method achieves the best overall performance on both missing and observed classes. All results of compared methods are based on our re-implementations.

| Method | $\gamma = 67\%$ $A_m$ | $A_o$ | $H$ | $\gamma = 33\%$ $A_m$ | $A_o$ | $H$ | $\gamma = 0\%$ $A_m$ | $A_o$ | $H$ |
|---|---|---|---|---|---|---|---|---|---|
| STL | $0.00 \pm 0.00$ | $\mathbf{97.99} \pm 0.12$ | $0.00 \pm 0.00$ | $0.00 \pm 0.00$ | $\mathbf{87.32} \pm 0.25$ | $0.00 \pm 0.00$ | - | $84.33 \pm 0.36$ | - |
| ERM | $8.74 \pm 0.42$ | $93.95 \pm 0.15$ | $15.16 \pm 0.24$ | $15.52 \pm 0.53$ | $84.24 \pm 0.16$ | $25.96 \pm 0.22$ | - | $83.48 \pm 0.49$ | - |
| PCGrad | $8.04 \pm 0.39$ | $91.62 \pm 0.21$ | $14.51 \pm 0.33$ | $14.28 \pm 0.47$ | $82.77 \pm 0.24$ | $23.53 \pm 0.35$ | - | $84.11 \pm 0.41$ | - |
| WeighLosses | $7.73 \pm 0.45$ | $89.68 \pm 0.29$ | $13.07 \pm 0.36$ | $14.25 \pm 0.39$ | $85.56 \pm 0.31$ | $24.35 \pm 0.34$ | - | $84.20 \pm 0.38$ | - |
| **Ours** | $\mathbf{10.82} \pm 0.38$ | $90.29 \pm 0.24$ | $\mathbf{18.17} \pm 0.31$ | $\mathbf{16.58} \pm 0.41$ | $86.62 \pm 0.28$ | $\mathbf{27.21} \pm 0.37$ | - | $\mathbf{85.98} \pm 0.43$ | - |

# F  Additional Results

## F.1  Benefits of the graph structure

To show the benefits of the graph structure, We conduct an experiment by directly incorporating the class knowledge into the instance with a cross attention module on `Office-Home` with a missing rate of 75%. ✓ and ✗ denote whether the model explores relationships between nodes or not.

In Table F.8, the direct incorporation obtains lower performance than our method using the association graph. This is because the direct incorporation does not capture relationships among nodes and therefore fails to fully utilize the structure information to transfer the task-specific and class-specific knowledge to each instance. We also found that both methods outperform single-task learning in terms of the harmonic mean (H). This again shows that our multi-task models benefit not only from the knowledge stored in the graph nodes, but also from the relationships among them.

Table F.8: Comparisons between with and without the graph structure.

| Methods | Graph structure | $A_m$ | $A_o$ | H |
|---|---|---|---|---|
| Single-task learning | ✗ | 0.00 | 88.25 | 0.00 |
| Direct incorporation | ✗ | 37.45 | 88.84 | 50.94 |
| Ours | ✓ | 47.51 | 87.16 | 60.59 |

## F.2  Compared with out-of-distribution generalization methods

We make comparisons to three typical out-of-distribution generalization methods under the proposed settings on `Office-Home`. As shown in Table F.9, our method outperforms them consistently, showing the effectiveness of our association graph learning in dealing with category shifts.

Table F.9: Comparisons between out-of-distribution methods and the proposed method.

| Missing rates | 75% | 50% | 25% | 0% |
|---|---|---|---|---|
| MixUp [10] | 51.18 | 61.06 | 64.38 | 81.53 |
| MixStyle [11] | 52.95 | 63.43 | 65.97 | 81.59 |
| IRM [1] | 57.11 | 64.11 | 66.24 | 81.41 |
| Ours | **60.59** | **65.54** | **67.12** | **82.01** |

## F.3 Computation cost

We calculate the computation cost in Table F.10 for each iteration (the unit for time is the second, experiments on Office-Home with missing rate 75%). The model with the association graph takes more time to test in each iteration than the model without the graph. The main reason is that the graph model needs to compute edges for pair-wise nodes in the graph. However, the model with the association graph significantly outperforms without the graph, by a large margin of 10.74%, in terms of the harmonic mean. Moreover, we also find that as the number of GNN layers increases, the inference time increases only slightly. Thus, we can conclude that the computational cost is mainly from the graph construction rather than GNN layers.

Table F.10: Comparisons on the computation cost during inference.

| Number of GNN layers | 1 | 2 | 3 | 4 | 5 |
|---|---|---|---|---|---|
| w/o Association graph | 0.14 | 0.14 | 0.14 | 0.14 | 0.14 |
| w Association graph | 0.68 | 0.69 | 0.71 | 0.72 | 0.73 |

## F.4 Benefits of each proposed architecture

We provide the ablation study to show the benefits of each sub-graphs in the proposed association graph. The results are shown in Table F.11. ✓ and ✗ denote whether the association graph contains the corresponding sub-graphs or not. From the table, we can see that the model containing the task and class graphs outperforms the model with task or class graphs. This demonstrates that our model benefits from each sub-graphs in the association graph.

Table F.11: Benefits of each sub-graphs in the proposed association graph.

| Task graph | Class graph | $A_m$ | $A_o$ | H |
|---|---|---|---|---|
| ✗ | ✓ | 46.21 | 86.10 | 59.34 |
| ✓ | ✗ | 45.80 | 84.02 | 58.41 |
| ✓ | ✓ | **47.51** | **87.16** | **60.59** |

## F.5 Advantages of using assignment entropy maximization

To show the advantage of using assignment entropy maximization, we set different values of $\beta$ for assignment entropy maximization in Table F.12 (on Office-Home with the missing rate of 75%). During training, $\beta$ controls the trade-off between the cross-entropy and assignment entropy losses. When $beta$ is 0, the assignment entropy maximization is not optimized during training. The model with $\beta \neq 0$ outperforms the model with $\beta = 0$ consistently in terms of the average accuracy of missing classes and the harmonic mean. This demonstrates that the assignment entropy maximization can improve the knowledge transferring for missing classes in the association graph, making the test instance more discriminative.

Table F.12: Comparisons between the models with the different values of $\beta$.

| $\beta$ | $A_m$ | $A_o$ | H |
|---|---|---|---|
| 0 | 44.65 | **87.59** | 58.26 |
| 1 | 46.12 | 87.02 | 59.30 |
| 0.1 | **47.51** | 87.16 | **60.59** |
| 0.01 | 46.33 | 87.11 | 59.63 |
| 0.001 | 45.59 | 86.90 | 58.96 |
| 0.0001 | 45.13 | 86.69 | 58.37 |

We further evaluate a variant with fixed and equal weights of edges from tasks to a class. As shown in Table F.13, our method outperforms the variant. This demonstrates that with the assignment entropy maximization, the learned weights between tasks and a class are not the same.

Table F.13: Comparisons between the variant with fixed and equal weights and the proposed method.

| Edge weights between each task and a class | $A_m$ | $A_o$ | H |
|---|---|---|---|
| Fixed and equal | 46.46 | 86.97 | 59.35 |
| Ours | **47.51** | **87.16** | **60.59** |

### F.6 Benefits of the designed metric function

We provide the results of a variant using the learnable metric function for the edge between task and class graphs in Table F.14. The results show that our method outperforms this variant.

Table F.14: Comparisons between the variant using the learnable metric function for the edge between task and the proposed method.

| Metric function | $A_m$ | $A_o$ | H |
|---|---|---|---|
| Learnable | 46.39 | 87.20 | 59.79 |
| Ours | 47.51 | 87.16 | 60.59 |

## G Discussion about Limitations

In this paper, the proposed setting is based on the multi-input multi-output setting for multi-task learning, where different tasks have different data distributions and share the same label or target spaces. One potential limitation of the work is that the proposed setting is not applicable to the single-input multi-output setting. The reason is that different tasks do not share the same label or target spaces in the single-input multi-output setting.

Moreover, based on the multi-input multi-output setting, we address category shifts in multi-task classification. Category shift means that the training label space is a subset of the test label space in each task, which only appears between several related classification tasks. Thus, the other limitation of the work is that our method is not directly applicable to regression tasks. Our work could be extended to other settings to explore and utilize the structural information. We leave the explorations for future work.

Our model is designed for finite classes and therefore is not directly applicable to infinite out-of-domain classes. A possible extension is to construct a placeholder node in the graph to represent all unknown open classes during training.

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

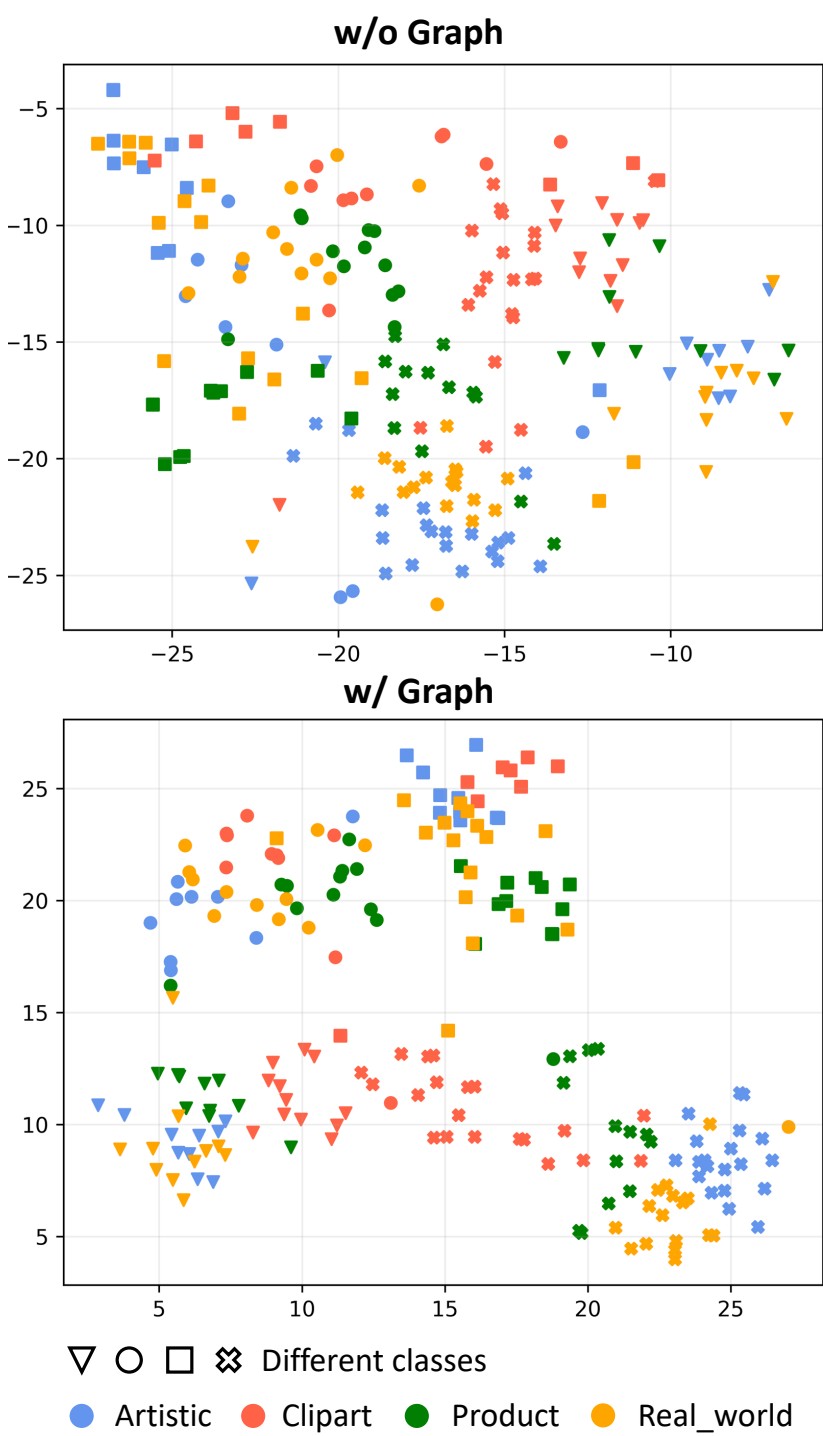

Figure E.1: Benefit of the association graph in feature learning. We show the visualizations of the test features without (top) and with the graph (bottom) from `Office-Home`. Different shapes denote different classes, and different colors correspond to different tasks. All classes in the figure are observed during training for `Artistic`, while missed for other tasks during training. Our graph encourages the features from the other tasks to be close to that from `Artistic`, which demonstrates the effectiveness of our model in generalizing the categorical information from seen classes (of `Artistic`) to missing classes (of other tasks).

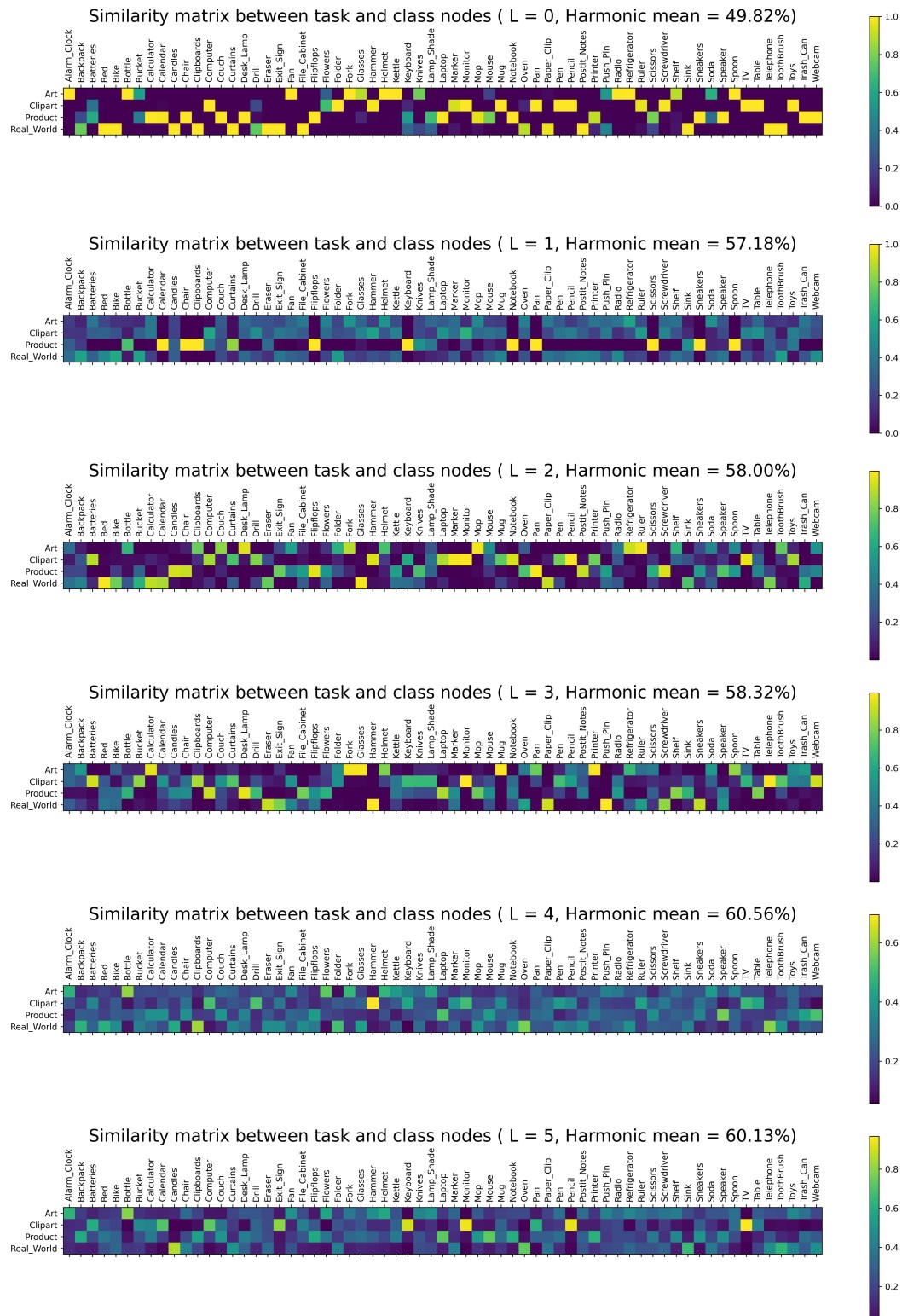

Figure E.2: Visualization of the relationships between task and classes.