# OpenReview forum: "Association Graph Learning for Multi-Task Classification with Category Shifts"
_NeurIPS.cc/2022/Conference — NeurIPS 2022 Accept_

### Official Review · Reviewer_n1qJ · 2022-07-08

**Rating:** 5
**Confidence:** 4
**Soundness:** 3 good
**Presentation:** 3 good
**Contribution:** 2 fair

**Summary:**

This article discusses the problems of category shifts in multi-task learning. To solve the problem, the author prposes to build a heterogeneous graph called association graph, where nodes represent tasks and classes and edges indicate their relationships. There are three kinds of edges: task-task, class-class, and task-class. The authors use the average of the corresponding sample representations to denote the tasks and classes. In addition, they propose three learnable metric functions to calculate the weights of the three kinds of relationships mentioned above. Intuitively, the edge weight of the relationship between the task and the class that does not appear in its training set should be low. Because the association graph is used for message passing, the authors hopes that the task can benefit from the information of unseen class nodes. To tackle the category shifts problems, they propose an entropy maximization mechanism to ensure that the relationships from a class to each task tend to be equal.  As a part of the loss functions, entropy maximization trains the model together with the classification loss of each task. In the testing phase, the instance node is inserted into the association graph by calculating the relationship with each task and each class. Afterwards, the instance representation is updated through message passing and then input to the classifier of the corresponding task to obtain the predicted label. To evaluate the proposed model, ablation experiments and visualization are conducted to show the competitive performance of the proposed model.

**Questions:**

1)	What is the specialty  of the designed metric function? Why the task graph and class graph need to be learnable to build edges, while the association graph does not use learnable to build edges between task and class? The author should discuss the motication and the difference behind these metric functions, rather than just show the formula.
2)	The associated graph is a heterogeneous graph. Have you considered or tried to use a heterogeneous GNN model. If so, how about the effect?


**Limitations:**

The authors described the limitations of their work and did not discuss any potential negative societal impacts of the work.

**Strengths And Weaknesses:**

Strengths
1)	This article explicitly defines the problem called category shifts in the multi-task classification.
2)	It proposed to address the category shifts problem with the association graph, which is a heterogeneous graph to describe the relationship between tasks and classes and enables tasks to benefit from the information propagated by unseen classes.
3)	The experiment part is relatively complete, where they have tested different parameters, and conducted ablation experiments. At the same time, they also provide the source code to ensure the reproducibility of the model.

Weaknesses
1)	Using assignment entropy as the part of loss function is far-fetched. It assumes that the weight of the edge from each task to a class tends to be the same in the association graph, that is, a class passes message to different tasks with equally importance. Obviously, it is not reasonable in practical application. As mentioned in the paper, due to patient populations or proprietary use of regulations, different tasks do not have the same diagnosis categories distribution, that also means, a class has different probabilities of appearance in different tasks. Therefore, the importance of tasks that the class appeared in training set should be higher than that of tasks that the class have not appeared in training set. However, the association graph trained by assignment entropy maximization assumes that each task should have the same importance to a class, which is against reality.
2)	The scenarios of category shifts problems are limited in the practical application. It requires the assumption that related tasks share the same label space in the testing space. In the evaluation part, the training data is fabricated by sample masking to simulate this scenario. Can the author give some examples in real life applications, such as giving some practical data and calculating their missing rate (definition 2)?
3)	The association graph is constructed as a small graph node with only about 70 nodes (Office-home dataset), but it requires 4 or even 6 layers of GNN for message passing? The association graph should be visualized to reveal the relationship between tasks and classes after training.
4)	Assignment entropy maximization is regarded as the objective to optimize the matching the association graph and multi-task, but the investigation is rather heuristic. In the ablation experiment (section of benefit of assignment entropy maximization), the advantages of using assignment entropy maximization should be evaluated.

---

> ### Author Response · Authors · 2022-08-02
> **Response to Reviewer n1qJ (2/2)**
>
> **Advantages of using assignment entropy maximization:**
> To show the advantages of using assignment entropy maximization, we provide two additional experiments:
>
> (1) We set different values of $\beta$ for assignment entropy maximization in the following table (on the Office-Home dataset with the missing rate of 75\%). During training, $\beta$ controls the trade-off between the cross-entropy and assignment entropy losses. When $beta$ is $0$, the assignment entropy maximization is not optimized during training.  The model with $\beta \neq 0$ outperforms the model with $\beta=0$ consistently in terms of the average accuracy of missing classes and the harmonic mean. This demonstrates that the assignment entropy maximization can improve the knowledge transfer for missing classes in the association graph, making the test instance more discriminative.
>
> | $\beta$ | $A_m$     | $A_o$     | $H$       |
> |---------|-----------|-----------|-----------|
> | 0       | 44.65     | **87.59** | 58.26     |
> | 1       | 46.12     | 87.02     | 59.30     |
> | 0.1     | **47.51** | 87.16     | **60.59** |
> | 0.01    | 46.33     | 87.11     | 59.63     |
> | 0.001   | 45.59     | 86.90     | 58.96     |
> | 0.0001  | 45.13     | 86.69     | 58.37     |
> |         |           |           |           |
>
> (2) We further evaluate a variant with fixed and equal weights of edges from tasks to a class. As shown in the table below, our method outperforms the variant. This demonstrates that with the assignment entropy maximization, the learned weights between tasks and a class are not the same. We added the discussions and all experimental results to the paper.
>
> | Edge weights between each task and a class | $A_m$     | $A_o$     | $H$        |
> |--------------------------------------------|-----------|-----------|-----------|
> | Fixed and equal                            | 46.46     | 86.97     | 59.35     |
> | Ours                                       | **47.51** | **87.16** | **60.59** |
> |                                            |           |           |           |
>
> **Designed metric functions and their motivations:**
> The specialty of the metric functions in the proposed graph is that metric functions are different in constructing edges of different types. This helps the model to control knowledge transfer among heterogeneous nodes for refinement and update of each instance.
> The differences and motivations of different metric functions are provided below:
>
> * We use the metric function with learnable parameters for the edges between homogeneous nodes, such as the edges in the task and class graphs. The motivation is to explore the task or class semantic information in the corresponding graph.
>
> * We use the fixed metric function for the edges between heterogeneous nodes, such as the edges between the task and class graphs. The motivation is to alleviate the overfitting of the learned connections between the task and its observed classes.
>
> For your interest, we further provide the results of a variant using the learnable metric function for the edge between task and class graphs in the following table. The results show that our method outperforms this variant. We added the discussions and experimental results to our paper.
>
> | Metric function for edges between task and class graphs | $A_m$ | $A_o$ | $H$ |
> |---------------------------------------------------------|-------|-------|-------|
> | Learnable                                               | 46.39 | 87.20 | 59.79 |
> | Ours                                                    | 47.51 | 87.16 | 60.59 |
> |                                                         |       |       |       |
>
> **Related works about the heterogeneous GNNs:**
> Thank you for this suggestion. Heterogeneous GNN models [a-d] are designed for the graph data (e.g., academic graph and review graph), which can not construct the graph from the non-graph training data. Thus, it is non-trivial to directly use the HetGNN-based methods to solve the category shifts in our scenario. We added the discussions in the related work of this paper.
>
> References:
>
> [a] [Zhang, Chuxu, et al. "Heterogeneous graph neural network." Proceedings of the 25th ACM SIGKDD international conference on knowledge discovery and data mining. 2019.](https://dl.acm.org/doi/abs/10.1145/3292500.3330961)
>
> [b] [Wang, Xiao, et al. "Heterogeneous graph attention network." The world wide web conference. 2019.](https://arxiv.org/abs/1903.07293)
>
> [c] [Hu, Ziniu, et al. "Heterogeneous graph transformer." Proceedings of The Web Conference 2020. 2020.](https://arxiv.org/abs/2003.01332)
>
> [d] [Fu, Xinyu, et al. "Magnn: Metapath aggregated graph neural network for heterogeneous graph embedding." Proceedings of The Web Conference 2020.](https://dl.acm.org/doi/abs/10.1145/3366423.3380297)
>
> *We hope that this rebuttal further solidifies your positive outlook on our work, and we are happy to discuss if you need any further clarifications. Thank you.*

---

> ### Author Response · Authors · 2022-08-02
> **Response to Reviewer n1qJ (1/2)**
>
> *We thank Reviewer n1qJ for the positive review and thorough feedback. We are glad that the reviewer found that the addressed problem is defined explicitly and the experiments are relatively complete.*
>
> **Assignment entropy maximization:**
> We regret the caused confusion here. The assignment entropy maximization is introduced as a regularizer to reduce spurious correlations.
> Due to the generalization gap between the training set with missing classes to the test set with complete classes, the model could learn spurious correlations between tasks and their corresponding observed classes in the given dataset. The spurious correlations will dominate the knowledge transfer between each task and its observed classes, hindering the missing classes from exploiting all categorical information.
>
> Without this regularization, the model tends to predict test instances as the observed classes. The assignment entropy maximization encourages the knowledge transfer between each task and its corresponding missing classes. This is also demonstrated by the ablation study in Table 1 of the paper, where the model with the regularization obtains higher accuracy in terms of the average accuracy of missing classes.
>
> Moreover, it is worth mentioning that the introduced hyperparameter $\beta$ controls the trade-off between the cross-entropy and assignment entropy losses. As a result, the weights of the edges from tasks to a class will not be forced to be exactly the same in the association graph. We added the discussions in our paper.
>
> **Practical applications:**
> The proposed setting is a new multi-task learning scenario.
> Its practical applications could not be limited by the mentioned assumption in the testing space. In our setting, the test label spaces of different tasks are not forced to be the same since the model predicts instances from different classes and tasks independently during inference. The real significance of this setting is coordinating multiple related tasks to make up for the missing information of each task from other tasks. In order to evaluate the methods under different degrees of category shifts, we set various missing rates of the training set.
>
> Many practical applications suffer from the category shift problem, for instance, fault diagnosis [11, 47], skin lesion classification [26, 52], and remote sensing image classification [32]. We give an example here [52] (https://people.ece.ubc.ca/bisicl/papers/cymiccai19.pdf). Some hospitals can only provide training data for limited diagnoses. In this case, the missing rate of this dataset is 40.82\%. The potential social impact of this setting is to expand the diagnostic scopes of each hospital for more comprehensive patient care, by collaborating all incomplete data from all hospitals. We added the discussions in our paper.
>
> **Stacked GNN layers:**
> It is necessary for our model to stack multiple GNN layers for sufficient knowledge transfer in the association graph. This is because the association graph is heterogeneous and the relationships between nodes can be very complex. By stacking multiple layers of GNNs, each node can eventually incorporate the knowledge from the large-hop neighbors across the entire graph to reduce the generalization gap between the training and test sets. This has been justified in Figure 3 of the paper: as the number of GNN layers (L) increases, our model performs increasingly better with the peak at L=4, which outperforms the one-layer of GNN by 3.38\% in terms of the harmonic mean. This discussion was added to our paper.
>
> **Visualization of the relationships between tasks and classes:**
> We visualize the similarity matrices between task and class nodes with different numbers of GNN layers in an anonymous figure (https://anonymous.4open.science/r/NeurIPS22_Submission6440/Visualization.pdf).  We can see that the model with the association graph (L>0) obtains more evenly distributed similarities than the model without the graph (L=0). Moreover, when L=4, fewer categories have absolutely dominated tasks (yellow square), which improves the knowledge transfer from observed classes to missing classes for each task. We add the visualization and discussions to the paper.

---

> > ### Comment · Reviewer_n1qJ · 2022-08-10
> > **Satisfied Clarifications**
> >
> > I appreciate the author's efforts to clarify my questions, I would like to raise my score to 5.

---

### Official Review · Reviewer_dn35 · 2022-07-08

**Rating:** 6
**Confidence:** 3
**Soundness:** 3 good
**Presentation:** 4 excellent
**Contribution:** 3 good

**Summary:**

The paper proposes a new problem setup where category shift exists in multi-task classification. The authors try to solve it by building graphs that represent tasks, instances and classes as nodes and assigning learnable weights among them. They also tried assignment entropy maximization to overcome the spurious features. The results show that using Graphsage on the proposed graph improve the performance significantly on several benchmarks under different category missing rate.


**Questions:**

-- How would a simple baseline of mixing everything together and training the model, work?

-- Have you tried using relational graph models?


**Limitations:**

Yes, the paper discusses potential negative societal impact of their work.

**Strengths And Weaknesses:**

++ The paper is well written and addresses an important problem.

++ The definition and representation of the graph are clear and easy to understand.

++ The idea for solving the spurious features (3.2) (to boose the connection of tasks with their unseen categories) is novel. The idea can be tried on other domain generalization tasks.

++ The performance is also consistent when a shift does not exist, which may imply a potential theoretical proof behind the method.

--The graph model may need some ablation study on the effectiveness of each proposed architecture.

-- The main limitation I am worried about is the extensibility of the model. It might be a little challenging for a graph model to expand especially since the out-of-domain classes can be theoretically infinite. In this case, the missing rate also does not make sense.

---

> ### Author Response · Authors · 2022-08-02
> **Response to Reviewer dn35**
>
> *We thank Reviewer dn35 for the positive review and thorough feedback. We are glad that the review found that the introduced regularization is novel, the performance is consistent, and the paper is well written and easy to understand.*
>
>
> **Effectiveness of each proposed architecture:**
> We provide the ablation study to show the benefits of each sub-graph in the proposed association graph. The results are shown in the following table. &radic; and  &times; denote whether the association graph contains the corresponding sub-graph or not. From the table, we can see that the model containing the task and class graphs outperforms the model with task or class graphs. This demonstrates that our model benefits from each sub-graphs in the association graph. We added this discussion and experimental results to our paper.
>
> | Task graph | Class graph | $A_m$     | $A_o$     | $H$       |
> |------------|:-------------:|-----------|-----------|-----------|
> | &times;     | &radic;     | 46.21     | 86.10     | 59.34     |
> | &radic;    | &times;      | 45.80     | 84.02     | 58.41     |
> | &radic;    | &radic;     | **47.51** | **87.16** | **60.59** |
> |            |             |           |           |           |
>
>
> **Extensibility of the model:**
>  Our model is designed for finite classes and therefore is not directly applicable to infinite out-of-domain classes. A possible extension is to construct a placeholder node in the graph to represent all unknown open classes during training. We leave the explorations for future work. The discussions were added to the paper.
>
> **A simple baseline:** The ERM [16] method is considered a simple baseline in the original version of our paper. Compared to the single-task learning baseline, ERM obtains better performance in terms of the harmonic mean, demonstrating that ERM enables knowledge sharing between tasks to some extent. However, on the Office-Home dataset with the 75\% missing rate, our model surpasses ERM by a large margin of 11.27\% (harmonic mean). The consistent improvements on all benchmarks with different missing rates further demonstrate that the association graph is effective in improving knowledge sharing for multi-task classification with category shifts. We made this clearer in the paper.
>
> **Related works on relational graph models:**
> Thank you for your mentioning this. Relational graph models [a-d] are relevant to our methods since both aim to fully utilize the structure information in the graph. However, the main difference is that most relational graph models are designed for the homogeneous graph, while the proposed graph handles heterogeneous nodes. It would be interesting to extend the module in the relational graph models in our association graph, such as the block-diagonal decomposition [a] and the hierarchical attention [c]. We leave the explorations for future work. The discussions and the related works were added to the paper.
>
> References:
>
> [a] [Schlichtkrull, Michael, et al. "Modeling relational data with graph convolutional networks." European semantic web conference. Springer, Cham, 2018.](https://link.springer.com/content/pdf/10.1007/978-3-319-93417-4.pdf)
>
> [b] [Vashishth, Shikhar, et al. "Composition-based multi-relational graph convolutional networks." International Conference on Learning Representations, 2020.](https://arxiv.org/pdf/1911.03082.pdf)
>
> [c] [Zhang, Zhao, et al. "Relational graph neural network with hierarchical attention for knowledge graph completion." Proceedings of the AAAI Conference on Artificial Intelligence. Vol. 34. No. 05. 2020.](https://ojs.aaai.org/index.php/AAAI/article/view/6508)
>
> [d] [Busbridge, Dan, et al. "Relational graph attention networks." arXiv preprint arXiv:1904.05811 (2019).](https://arxiv.org/pdf/1904.05811.pdf)
>
> *We hope that this rebuttal further solidifies your positive outlook on our work, and we are happy to discuss if you need any further clarifications. Thank you.*

---

> > ### Comment · Reviewer_dn35 · 2022-08-03
> > **Update after rebuttal**
> >
> >  I would like to thank the authors for a detailed rebuttal. I am convinced by the additional analyses that the authors performed. My score stays the same, i.e., a weak accept. I think it is a strong paper with a medium impact on the literature and no major concerns.

---

### Official Review · Reviewer_QjDy · 2022-07-09

**Rating:** 7
**Confidence:** 5
**Soundness:** 3 good
**Presentation:** 3 good
**Contribution:** 2 fair

**Summary:**

This paper aims to tackle the problem about the multi-task classification with category shifts. It proposes learning an association graph to transfer knowledge among tasks for missing classes, and constructs the association graph with nodes representing tasks, classes and instances. The proposed method encodes the relationships among the nodes in the edges to guide the knowledge transfer between them.  Moreover, to avoid spurious correlations between task and class nodes in the graph, the authors introduce an assignment entropy maximization that encourages each class node to balance its edge weights, which enables all tasks to fully utilize the categorical information from related tasks. An extensive evaluation on three general benchmarks and a medical dataset for skin lesion classification reveals that the proposed method consistently performs better than representative baselines.

**Questions:**

* The author should clarify the relationships between the proposed graph method and the heterogeneous GNNs.
* The author should clarify the relationships between the proposed setting and out-of-distribution (OOD) generalization problem, and add more OOD methods as baselines for fair comparisons.
* The author should add more related works involving above two areas.

**Limitations:**

The computational cost of the proposed method seems a lot, the authors do not provide necessary analysis and comparisons about it.

**Strengths And Weaknesses:**

**Strengths**
* This paper develops a new multi-task learning scenario, in which individual tasks do not contain complete training data for the categories in the test set.
* This paper utilize an association graph to propagate  and transfer the knowledge between the  constructed nodes, which is reasonable.

**Weaknesses**
* The construction and processing of the proposed association graph are similar to another research area-heterogeneous GNNs, and the author does not mention much about them.
* The proposed setting is a bit like a combination of multi-task learning and out-of-distribution generalization, which is of limited novelty.

---

> ### Author Response · Authors · 2022-08-02
> **Response to Reviewer QjDy (2/2)**
>
> **Computational cost:**
> We calculate the computation cost per iteration in the table below (the unit for time is the second, experiments on Office-Home with a missing rate of 75\%). From the table, we can see that the model with the association graph takes more time to test in each iteration than the model without the graph. The main reason is that the graph model needs to compute edges for pair-wise nodes in the graph. However, as shown in Table D.5 of the paper, the model with the association graph obtains better performance than the model without the graph, by a large margin of 10.74\%, in terms of the harmonic mean.
>
> Moreover, we also find that as the number of GNN layers increases, the inference time increases only slightly. Thus, we can conclude that the computational cost is mainly from the graph construction rather than GNN layers. We added the discussions and analysis in the paper.
>
> | Number of GNN layers | 1    | 2    | 3    | 4    | 5    |
> |-------------------------------|------|------|------|------|------|
> | w/o Association graph         | 0.14 | 0.14 | 0.14 | 0.14 | 0.14 |
> | w Association graph           | 0.68 | 0.69 | 0.71 | 0.72 | 0.73 |
> |                               |      |      |      |      |      |
>
> References:
>
> [a] [Zhang, Chuxu, et al. "Heterogeneous graph neural network." Proceedings of the 25th ACM SIGKDD international conference on knowledge discovery and data mining. 2019.](https://dl.acm.org/doi/abs/10.1145/3292500.3330961)
>
> [b] [Wang, Xiao, et al. "Heterogeneous graph attention network." The world wide web conference. 2019.](https://arxiv.org/abs/1903.07293)
>
> [c] [Hu, Ziniu, et al. "Heterogeneous graph transformer." Proceedings of The Web Conference 2020. 2020.](https://arxiv.org/abs/2003.01332)
>
> [d] [Fu, Xinyu, et al. "Magnn: Metapath aggregated graph neural network for heterogeneous graph embedding." Proceedings of The Web Conference 2020.](https://dl.acm.org/doi/abs/10.1145/3366423.3380297)
>
> [e] [Yun, Seongjun, et al. "Graph transformer networks." Advances in neural information processing systems 32 (2019).](https://arxiv.org/abs/1911.06455)
>
> [f] [Lv, Qingsong, et al. "Are we really making much progress? Revisiting, benchmarking and refining heterogeneous graph neural networks." Proceedings of the 27th ACM SIGKDD Conference on Knowledge Discovery and Data Mining. 2021.](https://dl.acm.org/doi/abs/10.1145/3447548.3467350)
>
> [g] [Shen, Zheyan, et al. "Towards out-of-distribution generalization: A survey." arXiv preprint arXiv:2108.13624 (2021).](https://arxiv.org/abs/2108.13624)
>
> [h] [Zhou, Kaiyang, et al. "Domain generalization: A survey." (2021).](https://arxiv.org/abs/2103.02503)
>
> [i] [Wang, Jindong, et al. "Generalizing to unseen domains: A survey on domain generalization." IEEE Transactions on Knowledge and Data Engineering (2022).](https://arxiv.org/abs/2103.03097)
>
> [j] [Zhang, Hongyi, et al. "mixup: Beyond empirical risk minimization." International Conference on Learning Representations (2017).](https://arxiv.org/abs/1710.09412)
>
> [k] [Zhou, Kaiyang, et al. "Domain generalization with mixstyle."  International Conference on Learning Representations (2021).](https://openreview.net/forum?id=6xHJ37MVxxp)
>
> [l] [Arjovsky, Martin, et al. "Invariant risk minimization." arXiv preprint arXiv:1907.02893 (2019).](https://arxiv.org/abs/1907.02893)
>
> *We hope that this rebuttal further solidifies your positive outlook on our work, and we are happy to discuss if you need any further clarifications. Thank you.*

---

> ### Author Response · Authors · 2022-08-02
> **Response to Reviewer QjDy (1/2)**
>
> *We thank Reviewer QjDy for the positive review and constructive feedback. We are glad that the review found that the developed setting is new and our method is reasonable.*
>
>
> **Related works on Heterogeneous GNNs:**
> Thank you for bringing up related research topics, which we have included in the related work of our paper for discussion.
> Though our work also deals with heterogeneous graphs as in Heterogeneous GNNs (HetGNNs) models [a-f], the problem settings and technical implementations are fundamentally different. We summarize the main differences below:
>
> * HetGNNs-based methods focus on graph data (e.g., academic graph and review graph). However, our method learns an association graph from the non-graph data.
>
> * Due to the structure and content heterogeneity of the graph data, HetGNN-based methods need pre-processing modules for each node type to encode heterogeneous contents as a fixed-size embedding.
> However, the association graph has different types of nodes in the same feature space and thus can directly apply GNNs to refine each node with all types of neighbors.
>
> * Most HetGNNs-based methods [a-d] depend on the hierarchical architecture to aggregate content embeddings of heterogeneous neighbors for each node, which contains node-level (intra-metapath) and semantic-level (inter-metapath) aggregations. However, our method enables pair-wise nodes to transfer knowledge whether they are from the same type or not. Thus, our method is suitable for solving the complex knowledge transfer in multi-task classification with category shifts.
>
> **Related works on out-of-distribution generalization:** We would like to take this opportunity to stress the relationships between out-of-distribution generalization [g-l] and our setting.
> As shown in the table below, we show the differences in the two major aspects:
>
> * In the input space, out-of-distribution generalization separates all domains as the source domain(s) and the target domain(s), while our setting observes all domains during the training and test. As a consequence, out-of-distribution generalization focuses on single-directional knowledge transfer only from the source to the target domains, while our setting aims to learn bi-directional knowledge transfer between pair-wise domains (tasks).
> * In the label space, out-of-distribution has the complete label space that is shared by all source and target domains. In contrast, our setting has incomplete training label space for each task, which leads to more complex knowledge transfer.
>
> Note: $L^d$ and $U^d$ denote the labeled and unlabeled distributions from the domain (task) $d$, while $\mathcal{Y}$ denotes the corresponding label space and $ i,j \in \{1, 2, ..., T\}$.
>
> | Setup                              | Training input space   | Test input space     | Training label spaces                       | Test label spaces                    |
> |------------------------------------|------------------------|----------------------|---------------------------------------------|--------------------------------------|
> | Out-of-distribution generalization | $L^1, \ldots, L^{T-1}$ | $U^{T}$              | $\mathcal{Y}^1= \ldots = \mathcal{Y}^{T-1}$ | $\mathcal{Y}^{T} = \mathcal{Y}^1$    |
> | Our setting (Missing rate $>0$)    | $L^1, \ldots, L^{T}$   | $U^1, \ldots, U^{T}$ | $\mathcal{Y}^i \neq \mathcal{Y}^{j}$        | $\mathcal{Y}^i \neq \mathcal{Y}^{j}$ |
> |                                    |                        |                      |                                             |                                      |
>
> For your interest, we make an experimental comparison to three representative out-of-distribution generalization methods (MixUp [j], MixStyle [k], and IRM [l]) under the proposed settings on the Office-Home dataset. As shown in the table below, our method outperforms them consistently, demonstrating the effectiveness of our association graph learning in dealing with category shifts. We added the discussions and experimental results to the paper. Thank you.
>
> | Missing rates | 75\%      | 50\%      | 25\%      | 0\%       |
> |---------------|-----------|-----------|-----------|-----------|
> | MixUp [j]     | 51.18     | 61.06     | 64.38     | 81.53     |
> | MixStyle [k]  | 52.95     | 63.43     | 65.97     | 81.59     |
> | IRM [l]       | 57.11     | 64.11     | 66.24     | 81.41     |
> | Ours          | **60.59** | **65.54** | **67.12** | **82.01** |
> |               |           |           |           |           |

---

> > ### Comment · Reviewer_QjDy · 2022-08-03
> > **More questions**
> >
> > 1. With respect to the form of exchanging information between heterogeneous nodes, could you please clarify the differences between your method and [1] in your revision?\
> > [1] Liu H, Yang Y. Cross-graph learning of multi-relational associations[C]//International Conference on Machine Learning. PMLR, 2016: 2235-2243.
> >
> > 2. The idea about constructing adjacency matrix of heterogeneous nodes is formally similar to some previous work [2][3], could you clarify the differences between your method and these works [2][3] at related work in your revision? \
> > [2] Hu Z, Dong Y, Wang K, et al. Heterogeneous graph transformer[C]//Proceedings of The Web Conference 2020. 2020: 2704-2710.\
> > [3] Yang L, Hong S. Omni-Granular Ego-Semantic Propagation for Self-Supervised Graph Representation Learning[C]//Proceedings of the 39th International Conference on Machine Learning, PMLR 162:25022-25037.
> >
> > 3. Looking forward to your responses to my questions

---

> > > ### Author Response · Authors · 2022-08-04
> > > **Further response to Reviewer QjDy**
> > >
> > > *Thank you for bringing up the interesting references. We have included all mentioned methods in the related work of the new revision for discussion. Thank you.*
> > >
> > > **Comparisons with [1]:**
> > >
> > > In the revision, we clarify the differences with respect to the form of the information exchanging as follows:
> > >
> > > * **Different technical modules.**
> > > [1] uses the spectral graph product and label propagation operators to exchange information between heterogeneous nodes, while our work performs message passing (e.g., GNNs).
> > >
> > > * **Different goals of exchanging information.**
> > > [1] aims to infer the unobserved multi-relations with observed multi-relations across the graphs. In contrast, our work aims to refine each instance node with the categorical information from heterogeneous nodes, which enables each instance to gain more discriminative and informative representations.
> > >
> > > **Comparisons with [2][3]:**
> > >
> > > In terms of the adjacency matrix between heterogeneous nodes, we clarify the main differences in the revision as follows:
> > >
> > > * **Different ways of graph construction.**
> > > [2] multiplies two adjacency matrices (edge types) of the heterogeneous graph to learn new meta-paths automatically.
> > > In contrast, our method constructs the connections between heterogeneous nodes by directly computing the similarities between any pair-wise nodes with different metric functions.
> > >
> > > * **Different metric functions.**
> > > [3] fixes the edges between heterogeneous nodes to one or zero. By contrast, our method constructs the connections between heterogeneous nodes through the similarity between these nodes.

---

> > > > ### Comment · Reviewer_QjDy · 2022-08-05
> > > > **Update after rebuttal**
> > > >
> > > > Thank you for detailed explanations, the authors have answered all my questions in the revision, and I have raised the score from 5 to 7.

---

### Official Review · Reviewer_XBgC · 2022-07-10

**Rating:** 6
**Confidence:** 3
**Soundness:** 2 fair
**Presentation:** 3 good
**Contribution:** 3 good

**Summary:**

The paper addresses the problem of multi-task classification with category shifts where training data for different tasks would not have same categories. To mitigate category shift problem, the authors construct association graph where different tasks, classes and instances are nodes and use message passing to transfer knowledge between them. Assignment entropy maximization is also designed to balance weights of edges. The experiments conducted on three general benchmarks and a realistic medical dataset for skin lesion classification demonstrate the effectiveness of the method.

**Questions:**

1. How does the graph structure help to transfer knowledge? I wonder what would happen if you directly use the class knowledge as extra information to help with prediction.
2. Considering results from Table 2, the best performance is achieved when the node can aggregate knowledge from all other nodes. Does that mean that the knowledge of each node can directly be incorporated into the input instance and local structure information of association graph is not important?

**Limitations:**

More experiments and explanations need to be included to show how the association graph structure information can help with the category drifts problem.

**Strengths And Weaknesses:**

Strengths:
1. Very well written paper, addressing category shifts problem in multi-task classification, which is realistic and interesting. The proposed method is also a reasonable way to mitigate this problem by constructing association graph and transferring knowledge, because the missing class for one task may appear in other tasks.
2. The paper is well developed and easy to follow. The proposed method is clearly presented.
3. Extensive experiments are conducted on both three general benchmarks and real-world datasets. Results show that the proposed method outperforms other baselines. Ablation tests are also conducted to demonstrate the effectiveness of association graph and assignment entropy maximization modules in the method.

Weaknesses:
1. More explanations need to be included to show the necessity of using graph to transfer knowledge. They can just combine all class knowledge directly with input instance instead of constructing a complex network. The benefit of association graph structure needs to be stated.
2. More experiments can also be added to deal with the problem stated above and show the benefit of association graph structure. For example, what happens if the knowledge of class is directly combined with input instance or the association graph is complete.

---

> ### Author Response · Authors · 2022-08-02
> **Response to Reviewer XBgC**
>
> *We thank Reviewer XBgC for the positive review and thorough feedback.
> We are glad that the reviewer found that the addressed problem is realistic and interesting, the proposed method is reasonable, and the paper is very well written and easy to follow.*
>
> **Explanations about the necessity and benefits of the graph and its structure:**
> Thank you for your suggestion.
> The necessity of using graphs is due to category shifts requiring knowledge transfer among tasks and classes, which varies across different classes.  In the association graph, heterogeneous nodes stores the task and class knowledge, which provides transferable knowledge for each instance. The graph structure encodes complex relationships among heterogeneous nodes, which guides the knowledge transfer among these heterogeneous nodes. By message passing, each instance is refined with the categorical information stored in the association graph. As a result, the instance will gain more discriminative and informative representations. We added this discussion to our paper.
>
> **Experiments on the direct incorporation:**
> Following your suggestion, we conduct an experiment to directly incorporate the class knowledge into the instance by a cross attention module on Office-Home with a missing rate of 75\%. &radic; and  &times; denote whether the model explores relationships between nodes or not.
>
> The results in the table below show that the direct incorporation obtains lower performance than our method using the association graph. This is because the direct incorporation does not capture relationships among nodes and therefore fails to fully utilize the structure information to transfer the task-specific and class-specific knowledge to each instance. We also found that both methods outperform single-task learning in terms of the harmonic mean (H). This again shows that our multi-task models benefit not only from the knowledge stored in the graph nodes but also from the relationships among them. We added this discussion and experimental results to the paper.
>
> | Methods               | Graph Structure | $A_m$  | $A_o$    | $H$    |
> |-----------------------|:-----------------:|--------|----------|--------|
> | Single-Task Learning  | &times;         | 0.00   |  88.25   |  0.00  |
> | Direct incorporation  | &times;         | 37.45  | 88.84    | 50.94  |
> | Ours                  | &radic;         | 47.51  |   87.16  | 60.59  |
> | | | | | |
>
> *We hope that this rebuttal further solidifies your positive outlook on our work, and we are happy to discuss if you need any further clarifications.*

---

> > ### Comment · Reviewer_XBgC · 2022-08-10
> > **Update after rebuttal**
> >
> > Thanks for your detailed response. I think it is a technically solid, moderate-to-high impact paper and I will keep my score.

---

### Author Response · Authors · 2022-08-02
**Summary of the updates in the new version  of our paper**

*We thank all AnonReviewers for their insightful reviews, thoughtful comments, and supportive suggestions. We thank all AnonReviewers for their time and would like to address all the concerns raised through this rebuttal. Here, we provide a summary of the updates made in the new version, as suggested by the reviewers.*

**Main manuscript:**
* We have added the discussions about the necessity and benefits of using the graph to transfer knowledge in Section 3.1.

* We have added the motivations of the designed metric functions for different types of edges in Section 3.1.

* We have added the discussions about the related works on heterogeneous GNNs, relational graph models, and out-of-distribution generalization in Section 4.

* We have added the discussions about the stacked GNN layers and the simple baseline ERM [16] in Section 5.


**Supplementary Materials:**

* We have added the discussions about the practical application of the setting in Section B.

* We have provided the visualization of the similarity matrices between task and class nodes with different numbers of GNN layers in Section E.

* We have made comparisons with the variant with the direct incorporation and added the related discussions in Section F.1.

* We have provided comparisons with three representative multi-task learning methods, including MixUp, MixStyle, and IRM, in Section F.2.

* We have added the analysis and experimental results about the computation cost of the model in Section F.3.

* We have provided an ablation study to show the benefits of each sub-graph in the association graph in Section F.4.

* We have added new ablation studies to show the advantages of using assignment entropy maximization in Section F.5.

* We have added the experiments to show the benefits of the designed metric function in Section F.6.

* We have added the discussions about the extensibility of our work in Section G.

---

### Meta-Review · Area_Chair_fhwh · 2022-08-26

**Recommendation:** Accept
**Confidence:** Less certain

**Metareview:**

This paper focuses on a variant of multi-task classification that it argues is new and of practical interest. In this setting, multiple tasks share a label space, such as in the case of a domain shift setting in which classes exist in multiple domains. The distinguishing feature of this setting is that not all classes have observed instances in all tasks. The paper proposes a graph-based approach, in which nodes representing tasks, classes, and instances are connected in an "association graph." A graph neural network is then learned over the association graph. Learnable metric functions weight the edges. Then representations of instances are classified using task-specific classifiers. Experiments on three benchmark tasks and a skin lesion task show that the proposed approach can consistently outperform both multi-task methods and sensible baselines like empirical risk minimization on all data.

The reviewers generally found the paper well-written, the new setting interesting and important to study, and the experimental evaluation thorough. During the review process, the authors added several other experiments and discussion justifying the use of a graph structure to effectively share knowledge across tasks.

**Award:**

No

---

### Decision · Program_Chairs · 2022-09-14

Accept